# MIMT: Masked Image Modeling Transformer for Video Compression

**Jinxi Xiang**[*], **Kuan Tian**[*], **Jun Zhang**[†], **Xiao Han, Wei Yang**
Tencent AI Lab, Shenzhen
{jinxixang,kuantian,junejzhang,haroldhan,willyang}@tencent.com

## Abstract

Deep learning video compression outperforms its hand-craft counterparts with enhanced flexibility and capacity. One key component of the learned video codec is the autoregressive entropy model conditioned on spatial and temporal priors. Operating autoregressive on raster scanning order naively treats the context as unidirectional. This is neither efficient nor optimal considering that conditional information probably locates at the end of the sequence. We thus introduce an entropy model based on a masked image modeling transformer (MIMT) to learn the spatial-temporal dependencies. Video frames are first encoded into sequences of tokens and then processed with the transformer encoder as priors. The transformer decoder learns the probability mass functions (PMFs) *conditioned* on the priors and masked inputs, and then it is capable of selecting optimal decoding orders without a fixed direction. During training, MIMT aims to predict the PMFs of randomly masked tokens by attending to tokens in all directions. This allows MIMT to capture the temporal dependencies from encoded priors and the spatial dependencies from the unmasked tokens, i.e., decoded tokens. At inference time, the model begins with generating PMFs of all masked tokens in parallel and then decodes the frame iteratively from the previously-selected decoded tokens (i.e., with high confidence). In addition, we improve the overall performance with more techniques, e.g., manifold conditional priors accumulating a long range of information, shifted window attention to reduce complexity. Extensive experiments demonstrate the proposed MIMT framework equipped with the new transformer entropy model achieves state-of-the-art performance on HEVC, UVG, and MCL-JCV datasets, generally outperforming the VVC in terms of PSNR and SSIM.

## 1 Introduction

Videos continue to grow exponentially as demand for various video applications increases on social media platforms and mobile devices. Traditional video compression codecs, such as HEVC and VVC, are still moving toward more efficient, hardware-friendly, and versatile. However, their framework still followed a hybrid coding framework that remained unchanged decades ago: spatial-temporal prediction coding plus transformation-based residual coding.

Neural video compression surged to outperform handcraft codecs by optimizing the rate-distortion loss in an end-to-end manner. One line of earlier work replaces traditional coding modules, including motion estimation, optical-flow-based warping, and residual coding modules with neural networks. Recently, residual coding has been proved to be suboptimal compared with context coding. Moreover, a pixel in frame $x_t$ is related to all pixels in the previously decoded frames $x_{<t}$ and pixels already decoded at $x_t$. Due to the huge space, it is impossible for traditional video codecs to explore the correlation between all rules using handcrafted rules explicitly.

Using the entropy model to exploit the spatial-temporal dependencies from the current and past decoded frames can vastly reduce data redundancies. The transformer is rising for computer vision tasks, including low-level image analysis. Inspired by the language-translation model, VCT

---

[*]equal contribution
[†]corresponding author

(Mentzer et al., 2022) for the first time uses a transformer as the conditional entropy model to predict the probability mass function (PMF) from the previous frames. VCT uses the estimated probability to losslessly compress the quantized latent feature map $\hat{y}_t$ without direct warping or residual coding modules. The better the transformer predicts the PMFs, the fewer bits are required for the video frames.

For VCT, the transformer decoder is an autoregressive model which regards video frames naively as sequences of tokens and decodes the current frame $y_t$ sequentially in a raster scanning order (i.e., token-by-token). We find this strategy neither optimal nor efficient, and thus, we propose a masked image modeling transformer (MIMT) using bidirectional attention. During training, MIMT aims to optimize a proxy task similar to the mask prediction in BERT (Devlin et al., 2018) and BEIT (Bao et al., 2021) to predict the PMFs of masked tokens. At inference, MIMT adopts a novel non-sequential autoregressive decoding method to predict the image in a few steps. Each step keeps the most confident (smallest entropy) token for the next iteration.

Our contributions are summarized as follows: (1) We design an entropy model based on a bi-directional transformer MIMT to compress the spatial-temporal redundancy in video frames. MIMT is trained on masked image modeling tasks. It can capture temporal information from past frames and spatial information from the decoded tokens at inference time. (2) More techniques are introduced to make our video compression model versatile. We employ manifold priors, including the recurrent latent prior, to accumulate an extended range of decoded frames. To further reduce MIMT complexity, we introduce alternating transformer layers with non-overlapping shifted window attention. (3) The proposed MIMT achieves state-of-the-art compression results with all these improvements. It generally outperforms the last H.266 (VTM) in terms of PSNR and SSIM. The bitrate saving over H.266 (VTM) is 29.6% on the UVG dataset in terms of PSNR.

## 2 RELATED WORK

**Video Compression.** Lu et al. (2019) developed the DVC model with all modules in the traditional hybrid video codec replaced by the network. DVC-Pro is proposed with a more advanced entropy model and deeper network (Lu et al., 2020). Agustsson et al. (2020) extended optical-flow-based estimation to a 3D transformation by adding a scale dimension. Hu et al. (2020) considered rate-distortion optimization when encoding motion vectors. In Lin et al. (2020), a single reference frame is extended to multiple reference frames. Yang et al. (2020) proposed a residual encoder and decoder based on RNN to exploit accumulated temporal information.

Deviated from the residual coding, DCVC (Li et al., 2021) employed contextual coding to compensate for the shortness of the residual coding scheme. Mentzer et al. (2022) proposed simplifying the "hand-craft" video compression network of explicit motion estimation, warp, and residual coding with a transformer-based temporal model. Contemporary work from Li et al. (2022) uses multiple modules, e.g., learnable quantization and parallel entropy model, to improve significantly the compression performance, which surpasses the latest VVC codec. AlphaVC (Shi et al., 2022) introduced several techniques, e.g., conditional I-frame and pixel-to-feature motion prediction, to effectively improve the rate-distortion performance.

**Masked Image Modeling.** Masked language modeling, first proposed in BERT (Devlin et al., 2018), has revolutionized the field of natural language processing, significantly when scaling to large datasets and huge models (Brown et al., 2020). The success in NLP has also been replicated in vision tasks by masking patches of pixels (He et al., 2022) or masking tokens generated by a pretrained dVAE (Bao et al., 2021; Xie et al., 2022). Recently, these works have also been extended to other domains to learn good representations for action recognition (Tong et al., 2022; Feichtenhofer et al., 2022), video prediction (Gupta et al., 2022), and image generation (Chang et al., 2022; Wu et al., 2022).

## 3 METHOD

As shown in Fig. 1, we encode a sequence of (RGB) video frames $\{x_t\}_{t=1}^{T}$ into latent tokens $\{y_t\}_{t=1}^{T}$, using a CNN-based image encoder. Next, we get a temporal sequence $\{\hat{y}_{t-1}, \ldots, \hat{y}_1\}$ from the decoded frames buffer. We use decoded sequence to compress $y_t$ with the transformer

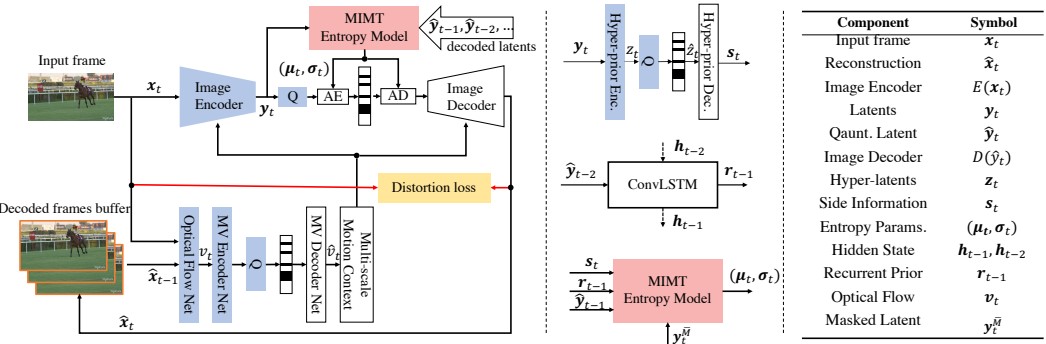

Figure 1: The core of the proposed video compression framework is a MIMT entropy model used to fully exploit the spatial-temporal correlation among video frames. The estimated parameters are generated from the side information prior $s_t$, recurrent prior $r_{t-1}$, and the last decoded latent $\hat{y}_{t-1}$.

entropy model for non-sequential autoregressive. At the receiver side, we recover $\hat{y}_t$ and reconstruct frames $\{\hat{x}_t\}_{t=1}^T$ with a CNN-based image decoder by feeding with the quantized latent tokens $\{\hat{y}_t\}_{t=1}^T$.

### 3.1 PRELIMINARY

Given the true PMF $p(y_t)$ and the estimated PMF $q(y_t)$, we use arithmetic coding (AC) (Langdon, 1984) to transmit $\{y_t\}_{t=1}^T$ with expected bit-rate of $H(p,q)$ which is expressed as the cross entropy:

$$H(p,q) = \mathbb{E}_{y_t \sim p}\left[-\log_2 q(y_t \mid c)\right], \tag{1}$$

with $c$ as the conditional information of $y_t$, e.g., hyper-prior, spatial autoregressive context.

The expected bit-rate can be reduced if $y_t$ has higher certainty. We can encode more frequently occurring values with fewer bits, and hence improve the efficiency. For this purpose, video coding exploits temporal relationship from previous frames $\{y_{t-1}, \ldots, y_1\}$ to reduce the estimated bit-rate:

$$H(p,q) = \mathbb{E}_{y_t \sim p}\left[-\log_2 q(y_t \mid y_{t-1}, \ldots, y_1, c)\right]. \tag{2}$$

Our main idea is to parameterize $q(y_t \mid y_{t-1}, \ldots, y_1, c)$ as a conditional distribution using a MIMT to minimize the cross-entropy $H(p,q)$.

### 3.2 CNN-BASED IMAGE ENCODER/DECODER

Video is transformed with a CNN-based image encoder to reduce dimensionality. Each frame $x_t \in \mathbb{R}^{H \times W \times 3}$ is represented as a grid of tokens $y_t \in \mathbb{R}^{h \times w \times d_C}$. The image encoder downsamples the raw image spatially and increases the channel dimension, yielding $(h, w, d_C)$ dimensional tokens where $(h, w)$ are $16\times$ smaller than $(H, W)$. We reconstruct image $\hat{x}_t$ with a decoder $D(\hat{y}_t)$.

One straightforward way is to train $E, D$ using standard neural image compression. But this *independent* transformation without any temporal hints would lay heavy burdens on the conditional coding transformer, considering that the previously decoded frame $\hat{x}_{t-1}$ loses rich information as it only contain 3 channels. Thus, it is also not optimal to learn temporal contexts merely from $\hat{x}_{t-1}$. Following the contextual encoder-decoder (Sheng et al., 2021; Li et al., 2022), we also build the image encoder $E$ and decoder $D$ using motion $v_t$ with little bit-stream overhead:

$$y_t = E(x_t \mid f_{\text{context}}(\hat{y}_{t-1}, \hat{v}_t)), \ \hat{x}_t = D(\hat{y}_t \mid f_{\text{context}}(\hat{y}_{t-1}, \hat{v}_t)) \tag{3}$$

where $v_t$ is estimated from optical flow network $f_{\text{mv}}(x_t, x_{t-1})$, and $f_{\text{context}}(\hat{y}_{t-1}, \hat{v}_t))$ generates multi-scale warped feature map *aligned* in the $t$-frame space.

### 3.3 MASKED IMAGE MODELING TRANSFORMER ENTROPY MODEL

Compressing the current frame $y_t$ can be formulated as a sequential autoregressive. Let $y_t = \left[y_t^i\right]_{i=1}^N$ denote the latent tokens obtained by feeding the image $x_t$ to the encoder, where $N$ is the

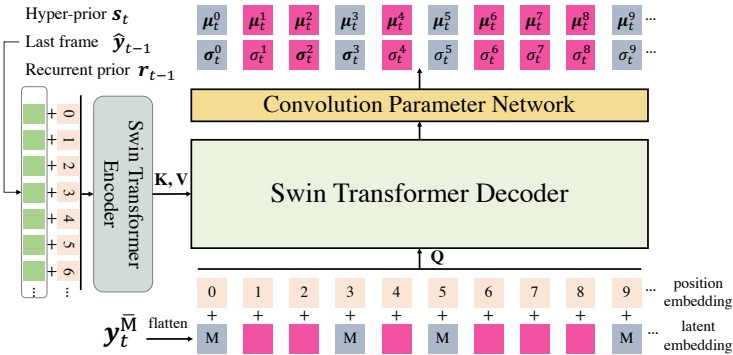

Figure 2: MIMT overview. During training, some of the tokens are masked, and the task is to predict the PMFs of masked tokens. At inference, the model begins with predicting all masked tokens and keeping the most certain ones, which are put back in sequence for the next prediction. We do this iteratively until all tokens are decoded.

length of the flattened token matrix. Given already transmitted tokens $\boldsymbol{y}_t^{<i}$, the transformer learns to predict the probability distribution of next index, i.e., $q(\boldsymbol{y}_t^i)$. This allows us to directly minimize:

$$R_{\text{AR}}(\boldsymbol{y}_t) = \mathbb{E}_{\boldsymbol{y}_t \sim p} \left[ -\sum_{i=1}^{N} \log_2 q\left(\boldsymbol{y}_t^i \mid \boldsymbol{y}_{t-1}, \ldots, \boldsymbol{y}_1, \boldsymbol{y}_t^{<i}\right) \right]. \tag{4}$$

As illustrated in Fig. 3, sequential autoregressive model can only make predictions based on the observed pixels (left upper part of the target pixel) due to the inductive bias caused by the strict adherence to the unidirectional scanning order. The fixed scanning order may achieve suboptimal performance. For example, if the informative condition is located at the end of the autoregressive sequence, it is difficult for the model to take full advantage of such relevant information.

We propose a bi-directional non-sequential transformer entropy model. The training task of the transformer is masked image modeling. The strategy is mask-then-predict which is straightforward: during training, we randomly mask some proportion of image tokens and feed the corrupted sequence to the transformer. Fig. 2 shows the overview of our transformer entropy model. $\mathbf{M} = [m_i]_{i=1}^{N}$ is the corresponding mask. We randomly sample a subset ($0\% \sim 100\%$) of tokens and replace them with a special learnable [M] token. The token $\boldsymbol{y}_t^i$ is replaced with [M] if $m_i = 1$, otherwise, if $m_i = 0$, $\boldsymbol{y}_t^i$ will be left intact.

Denote $\boldsymbol{y}_t^{\overline{\mathbf{M}}}$ as the masked sequence by applying $\mathbf{M}$ to $\boldsymbol{y}_t$. The optimization objective is to minimize the cross entropy of the predicted PMFs and the true distribution, i.e., the expected bit rate:

$$R_{\text{mask}}(\boldsymbol{y}_t) = \mathbb{E}_{\boldsymbol{y}_t \sim p} \left[ -\log_2 q\left(\boldsymbol{y}_t \mid \boldsymbol{y}_{t-1}, \ldots, \boldsymbol{y}_1, \boldsymbol{y}_t^{\overline{\mathbf{M}}}\right) \right]. \tag{5}$$

We aim to fully exploit the temporal correlations from video frames $\{\boldsymbol{y}_{t-1}, \ldots, \boldsymbol{y}_1\}$. In concrete, we feed manifold inputs to the MIM transformer and let the model learn complementary information from the rich temporal/spatial inputs which have different characteristics. Side information $\boldsymbol{s}_t$ is decoded from $\hat{\boldsymbol{z}}_t$ which is a hierarchical spatial estimation of $\boldsymbol{y}_t$; $\hat{\boldsymbol{y}}_{t-1}$ is most correlated latent with $\boldsymbol{y}_t$ in the past; $\boldsymbol{r}_{t-1}$ is from a recurrent network with convolution layers and ConvLSTM cells. Due to the recurrent structure, $\boldsymbol{r}_{t-1}$ is generated based on all previous latents $\{\hat{\boldsymbol{y}}_{t-2}, \ldots, \hat{\boldsymbol{y}}_1\}$.

After transformer encoding, we feed the rich encoded priors into a multi-layer bidirectional transformer to predict the PMFs of the masked tokens, where the cross-entropy between the ground-truth distribution and the predicted distribution is

$$R_{\text{mask}}(\boldsymbol{y}_t) = \mathbb{E}_{\boldsymbol{y}_t \sim p} \left[ -\log_2 q\left(\boldsymbol{y}_t \mid \boldsymbol{s}_t, \hat{\boldsymbol{y}}_{t-1}, \boldsymbol{r}_{t-1}, \boldsymbol{y}_t^{\overline{\mathbf{M}}}\right) \right]. \tag{6}$$

Notice the key difference to autoregressive modeling: the conditional dependency in MIMT is bidirectional that allows PMF prediction to utilize richer contexts in the current frame. For decoding,

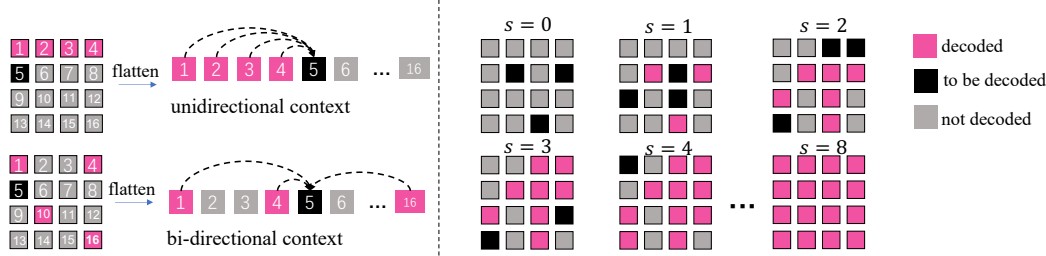

Figure 3: Left: Compared to the autoregressive model with a unidirectional context, MIM transformer benefits from the bi-directional context information. Right: With a pre-trained MIM transformer, we can decode $\boldsymbol{y}_t$ in a few steps. Each step recovers a portion of tokens based on the decoded ones. Only the most certain tokens with the largest entropy are kept for the next iteration.

PMFs of tokens are iteratively predicted with the MIM transformer following a decoded ratio scheduler at each step as shown below.

### 3.4  ITERATIVE DECODING SCHEDULER

At receiver side, we first decode the optical flow $\hat{\boldsymbol{v}}_t$ and the hyper-prior $\hat{\boldsymbol{z}}_t$ before decoding the latent $\hat{\boldsymbol{y}}_t$. Since we do not make any assumptions about the distribution of $\hat{\boldsymbol{v}}_t$ and $\hat{\boldsymbol{z}}_t$, a non-parametric, fully factorized entropy model is used for decoding (Ballé et al., 2018).

With the pre-trained MIMT, we can decode the current frame $\boldsymbol{y}_t$ in one step assuming all tokens in $\boldsymbol{y}_t^{\overline{\mathbf{M}}}$ are [M] token. But the PMFs obtained from all [M] tokens are not optimal estimations because the spatial context information in the current frame is totally ignored. Thus it would introduce a great bit-rate cost for arithmetic coding. Instead, we decompose the one-step decoding into several iterations in Fig. 3.

Let $\gamma(i) = \sin(i \cdot \frac{\pi}{2})$, where $i \in \{0, \frac{1}{n}, \ldots, \frac{n-1}{n}, 1\}$, be a decoded ratio scheduler that determines the number of tokens to be recovered at each step. $n = 8$ by default. The scheduler is monotonically increasing and $\gamma(0) \to 0$, $\gamma(1) \to 1$. At $i = 0$, we start with all tokens in $\boldsymbol{y}_t^{\overline{\mathbf{M}}}$ are [M]. Only the most certain $\lfloor \gamma(1) \cdot N \rceil$ tokens with the smallest entropy are kept for the next iteration. The entropy of a token is accumulated along the channel:

---

**Algorithm 1:** MIMT Iterative Decoding

**Input:** $\hat{\boldsymbol{y}}_{t-1}, \boldsymbol{r}_{t-1}, \gamma(i), n, \mathrm{Bs}$ ;  `// Bs is the bitstream.  n is step.`
**Output:** $\hat{\boldsymbol{x}}_t$ ;  `// reconstructed image`
`/* Initialization                 */`
Decode $\hat{\boldsymbol{v}}_t$ from Bs using arithmetic decoder ;
Decode $\hat{\boldsymbol{z}}_t$ from Bs using arithmetic decoder ;
Get $\boldsymbol{s}_t$ by pass $\hat{\boldsymbol{z}}_t$ to hyper-prior decoder;
$i \leftarrow 0, \mathbf{M} \leftarrow \mathbf{1}$ ;
`/* iterative decoding              */`
**while** $i \leq n$ **do**
$\quad (\hat{\boldsymbol{\mu}}_t, \hat{\boldsymbol{\sigma}}_t) \leftarrow \mathrm{MIMT}(\boldsymbol{y}_t^{\overline{\mathbf{M}}}, \boldsymbol{r}_{t-1}, \boldsymbol{s}_t, \hat{\boldsymbol{y}}_{t-1})$ ;
$\quad i \leftarrow i + 1$ ;
$\quad$ Select $\lfloor \gamma(i)N - \gamma(i-1)N \rceil$ tokens by (7);
$\quad$ Decode the selected tokens from Bs with $(\hat{\boldsymbol{\mu}}_t, \hat{\boldsymbol{\sigma}}_t)$;
$\quad$ Replace the masked token [M] in $\boldsymbol{y}_t^{\overline{\mathbf{M}}}$ with the decoded tokens ;
$\boldsymbol{r}_t \leftarrow R(\boldsymbol{r}_{t-1}, \boldsymbol{h}_{t-1})$ ;  `// recurrent net`
$\hat{\boldsymbol{y}}_t \leftarrow \boldsymbol{y}_t^{\overline{\mathbf{M}}}$ ;  `                // M = 0`
$\hat{\boldsymbol{x}}_t \leftarrow D(\hat{\boldsymbol{y}}_t \mid f_{\mathrm{context}}(\hat{\boldsymbol{y}}_{t-1}, \hat{\boldsymbol{v}}_t))$ ;

---

$$H(\hat{\boldsymbol{y}}_t^i) = -\log_2 \prod_{j=1}^{d_C} p(\mu_t^{i,j}), \;\; p(\mu_t^{i,j}) = c_t^{i,j}(\mu_t^{i,j} + \frac{1}{2}) - c_t^{i,j}(\mu_t^{i,j} - \frac{1}{2}) \tag{7}$$

where $d_C$ is dimension of tokens; $c_t^{i,j}(\cdot)$ is the cumulative function of Gaussian distribution $\mathcal{N}\left(\mu_t^{i,j}, \sigma_t^{i,j\,2}\right)$.

We iteratively replenish $\boldsymbol{y}_t^{\overline{\mathbf{M}}}$ with the decoded tokens until all tokens being decoded. The scheduler is consistent with the intuition that video decoding follows a flow of information from coarse to fine. In the beginning, the model decodes the "sketch" of the image with small entropy. The "sketch" is

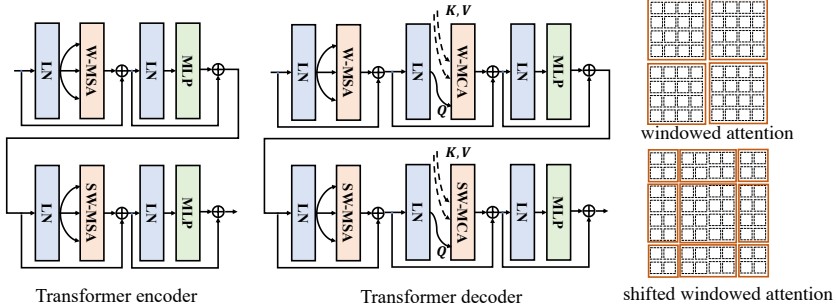

Figure 4: Transformer encoder and decoder with shifted-window attention. W-MSA: windowed multi-head self-attention. SW-MSA: shifted windowed multi-head self-attention. W-MCA: windowed multi-head cross attention. SW-MCA: shifted windowed multi-head cross attention.

easy to predict from frame redundancies. Gradually, with more decoded context, the model takes small steps to tweak the details considering the slow slop of `sine` around 1.

### 3.5 IMPLEMENTATION DETAILS

**Shifted window attention.** The global self-attention of the transformer is computationally unaffordable for video compression with long sequences. We use shifted window attention Liu et al. (2021) in both transformer encoder and decoder to alleviate this problem, as visualized in Fig. 4. The transformer encoder makes use of an interchange of shifted windows and thus, the computational cost is greatly reduced since self-attention is considered within local windows. In contrast to SW-MSA which uses the same input as the *key*, *query* and *value*, SW-MCA uses the encoded priors of $\{s_t, \hat{y}_{t-1}, r_{t-1}\}$ as the *key* and *value*, while using $y_t^{\overline{M}}$ as the *query*.

**Optimization loss.** Rate-distortion loss is optimized throughout the training process:

$$\mathcal{L}_{\text{RD}} = \underbrace{R(\hat{y}_t)}_{\text{rate (latents)}} + \underbrace{R(\hat{z}_t)}_{\text{rate (hyper-latents)}} + \underbrace{R(\hat{v}_t)}_{\text{rate (optical flow)}} + \lambda \cdot \underbrace{d(x_t - \hat{x}_t)}_{\text{distortion}}, \tag{8}$$

where $d(\cdot)$ represents the mean square error or MS-SSIM; $\lambda$ is a hyperparameter used to control the rate-distortion trade-off. We train four models with different $\lambda$ values $\{256, 512, 1024, 2048\}$. By default, we train models with MSE loss. When using the MS-SSIM metric, the model is fine-tuned with the MS-SSIM loss. *More training details are described in the Appendix.*

## 4 EXPERIMENTS

**Dataset.** We use Vimeo-90k (Xue et al., 2019) for training. The videos are randomly cropped into $256 \times 256$ patches. The test videos include HEVC Class B, UVG (Mercat et al., 2020), and MCL-JCV (Wang et al., 2016) datasets. All three datasets have a resolution of $1920 \times 1080$.

**Baselines.** We compare MIMT with widely acknowledged baselines, e.g., **x265**, **HM**, **VTM** codecs. x265 is implemented with the commercial FFmpeg, whereas HM and VTM are standard versions that can achieve much better performance but are extremely slow. We further obtains reported results from the following learned codecs: **DVCPro** (Lu et al., 2020), **FVC** (Hu et al., 2021), **DCVC** (Li et al., 2021), **C2F** (Hu et al., 2022), **VCT** (Mentzer et al., 2022), and **DMC** (Li et al., 2022).

**Metric.** The common PSNR and MS-SSIM are evaluated for all models(Wang et al., 2004) in RGB.

we set the GoP size as 32 for all datasets and use learned model (Cheng et al., 2020) for I frame compression. We center crop 1080p image into $1790 \times 1024$ to make it divisible by 256.

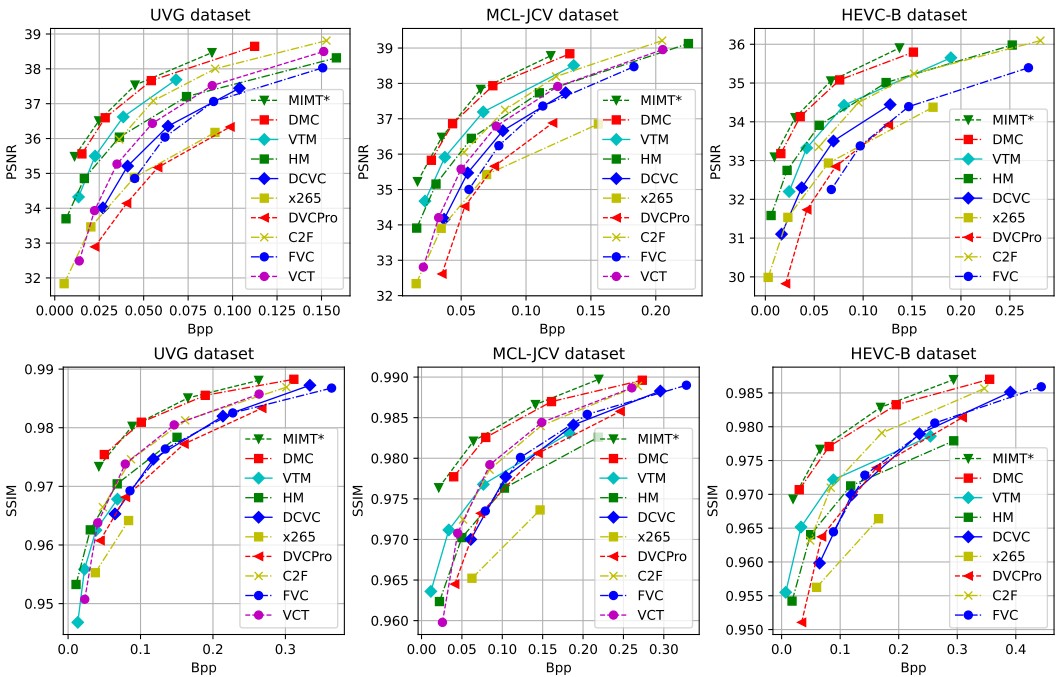

Figure 5: RD performance on UVG, MCL-JCV, and HEVC-B dataset.

|  | DVCPro | FVC | DCVC | HM | C2F | DMC | **MIMT** |
|---|---|---|---|---|---|---|---|
| UVG | +227.0% | +108.6% | +95.8% | +40.5% | +16.0% | −18.2% | −29.6% |
| MCL-JCV | +180.8% | +71.0% | +66.1% | +45.4% | +19.6% | −6.4% | −13.0% |
| HEVC-B | +209.8% | +108.3% | +49.8% | +40.4% | +14.2% | −5.1% | −25.5% |
| Average | +205.8% | +95.9% | +70.5% | +42.1% | +16.6% | −9.9% | −22.7% |

Table 1: BD-rate calculated by **PSNR** with respect to the anchor VTM.

# 5 RESULTS

## 5.1 RATE-DISTORTION PERFORMANCE

In Fig. 5, we plot the rate-distortion of our method and the baseline methods introduced in Section 4. We see that our MIMT outperforms most of these baselines across different datasets in terms of PSNR and SSIM. DMC is the best-performed competing method with a hybrid spatial-temporal entropy model. For MIMT, higher quality can be obtained with less bit rate cost under the same quality. This verifies the effectiveness of our entropy model in exploiting the temporal redundancies among video frames as well as spatial correlations among intra-frame tokens.

As shown in Table 1 and 2, we make quantitative metric with BD-rate Bjontegaard (2001) computed from PSNR-BPP and SSIM-BPP, respectively. The best traditional codec VTM is used as the anchor. From Table 1, we can find that DMC Li et al. (2022) is the only baseline method that surpasses VTM with 9.9% bite-rate saving. Our MIMT improves this performance further and it achieves an average of 22.7% bitrate saving over VTM on all datasets. From the perspective of SSIM, the improvement of MIMT is even larger, which is 56.3% bit rate saving over VTM. We notice that C2F and DMC also outperform VTM in terms of MS-SSIM.

## 5.2 ABLATION STUDIES

We conduct a series of ablation studies to demonstrate the effectiveness of different components.

**Entropy coding.** The MIMT uses a dynamic strategy to determine the encoding/decoding order for auto-regressive. Other strategies would be a raster scan order autoregressive entropy model like

|         | DVCPro  | FVC     | DCVC    | HM      | C2F     | DMC     | **MIMT** |
|---------|---------|---------|---------|---------|---------|---------|----------|
| UVG     | +68.1%  | +10.1%  | +33.6%  | +36.9%  | −20.1%  | −35.1%  | −40.0%   |
| MCL-JCV | +37.8%  | +15.0%  | +4.7%   | +43.7%  | −9.1%   | −46.8%  | −64.7%   |
| HEVC-B  | +61.7%  | +60.9%  | +31.0%  | +36.7%  | +12.7%  | −48.1%  | −64.2%   |
| Average | +55.8%  | +28.6%  | +23.1%  | +39.1%  | −5.5%   | −43.3%  | −56.3%   |

Table 2: BD-rate calculated by **SSIM** with respect to the anchor VTM.

| Modules | Ablation Option | **UVG** | **MCL-JCV** | **HEVC-B** | **Average** |
|---------|-----------------|---------|-------------|------------|-------------|
| MIMT | auto-regressive (*full*) | +5.2% | +6.9% | +7.1% | +6.4% |
| MIMT | auto-regressive (*block*) | +16.6% | +19.3% | +22.8% | +19.5% |
| MIMT | checkerboard | +15.1% | +17.7% | +19.2% | +17.3% |
| contextual encoder/decoder | non-contextual encoder/decoder | +13.0% | +18.4% | +19.5% | +16.9% |

Table 3: Ablations on the entropy model and the contextual encoder/decoder. The first column is the original design and the second column is the optional modules. The last four columns show the BD rate increase over the anchor of the complete MIMT model.

VCT (Mentzer et al., 2022), or a parallel-friendly checkerboard entropy model (He et al., 2021). For fair comparisons, both auto-regressive and checkerboard models use the same transformer encoder-decoder network, but they train and predict differently. For the auto-regressive model, we train the transformer with a self-attention mask to ensure causality similar to Vaswani et al. (2017), and then we predict each token one by one in raster scan order. We make two variants of auto-regressive: the *full* variant means operating on all $16 \times 16$ tokens, and the *block* variant means splitting the tokens into $4 \times 4$ non-overlapping blocks similar to VCT. No windowed attention mask is applied for autoregressive (*block*). As for the checkerboard, we predict the first half tokens with all input tokens masked, and then use the decoded tokens to predict the rest.

In Table 3, we calculate the BD-rate increase after ablations. In the first row, the MIMT entropy performs better than the *full* auto-regressive with +6.4% bitrate savings. The time-cost of auto-regressive (*full*) decoding is prohibitive for practical usage. As an alternative, auto-regressive (*block*) partitions the tokens into sub-blocks and decodes them in parallel. Although it can speed up greatly, it builds on the assumption that all blocks are independent, which is not necessarily true as concluded by VCT (Mentzer et al., 2022). In this way, the performance of autoregressive (*block*) degrades obviously with +19.5% bitrate increase. The parallel checkerboard entropy model is slightly better but still has a +17.3% increase. This is intuitively reasonable because the checkerboard model could be considered a simplified case of MIMT with fixed encoding/decoding order.

We follow the good practice to use a flow-based transformation coding (Sheng et al., 2021; Li et al., 2021; 2022). In the last row of Table 3, we conduct an ablation study on the image encoder by removing the temporal context. The results show the contextual encoder provides significant bitrate savings of +16.9% over the non-contextual encoder with independent image compression.

**Decoding scheduler**. The masked scheduler is a non-trivial design for encoding/decoding. The default scheduler is a `sine` function. We further explore two alternatives, i.e., `linear`, and `exponential` functions. The *sine* is concave, whereas the exponential is convex. A `sine` scheduler is in line with such intuitions: due to the temporal redundancy, the current frame has high similarities with previous frames. In the beginning, we can take a large step with high certainty. In the last few steps, we have to cope with pixels of large entropy

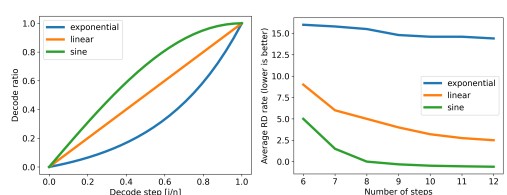

Figure 6: Decoding scheduler.

where we should take small steps to make more precise estimations. We adjust the number of steps from 6 to 12. As shown in Fig. 6, the results demonstrate that the `sine` performs consistently better than the others. The `linear` function provides decent results but is inferior to the `sine`.

The `exponential` performs poorly. *We provided further visualization in the Appendix, Section C*. Increasing the number of steps can improve the performance, but for steps larger than 8, the gains are marginal and it will introduce computational overhead for training and inference.

| hyper-prior $s_t$ | last frame $\hat{y}_{t-1}$ | recurrent prior $r_{t-1}$ | UVG | MCL-JCV | HEVC-B | Average |
|:---:|:---:|:---:|:---:|:---:|:---:|:---:|
| ✓ | ✓ | ✗ | +8.6% | +11.3% | +13.5% | +11.1% |
| ✓ | ✗ | ✓ | +9.2% | +14.9% | +15.6% | +13.2% |
| ✗ | ✓ | ✓ | +11.7% | +17.3% | +16.6% | +15.2% |
| ✓ | ✓ | ✓ | 0% | 0% | 0% | 0% |

Table 4: Ablations on the manifold inputs of MIMT. The figure in the table denotes the BD-rate increase over the full model used as the anchor.

**Manifold inputs.** MIMT takes three different kinds of inputs: side information $s_t$ from hyper prior, temporal prior from the last decoded frame $\hat{y}_{t-1}$, and temporal recurrent prior $r_{t-1}$. We conduct ablation studies to verify these priors for entropy modeling. As shown in Table 4, hyper-prior $s_t$ providing hierarchical information brings the most significant BD-rate improvements of +15.2%. $\hat{y}_{t-1}$ is the most temporally correlated frame in the past and it achieves +13.2% gains. By accumulating a long range of temporal information, the recurrent prior $r_{t-1}$ can effectively compensate for the overall performance with +11.1% BD-rate.

## 5.3 DECODING EFFICIENCY

Table 5 compares the model complexity in the number of parameters, MACs (multiply-accumulate operations), encoding time, and decoding time with learned codecs. The test video frames are 1080p and we use the publicly available source code of DVC, SSF, DCVC, and DMC. We deploy all models on a server with an NVIDIA P40 GPU. The time consumption includes bit-stream writing and reading with arithmetic coding. Note that all models are not optimized for coding efficiency.

In Table 5, we find that the decoding speed of DVC is moderate even without a sequential/recurrent network. DCVC is high of high complexity for encoding/decoding because it employs a sequential autoregressive context model. DMC splits the latents along the channel-wise and spatial-wise directions to accelerate the autoregressive. The proposed MIMT equipped with shifted window attention and non-sequential decoding strategy is comparable with DMC, in terms of model complexity and encoding and decoding time. For the other three variants of MIMT, the number of parameters is the same but they are trained differently, thus leading to varying MACs requirements. Not surprisingly, the parallel-friendly checkerboard is the most efficient consuming only 798 ms for decoding. The *full* autoregressive model is prohibitive slow at decoding, and the *block* auto-regressive reduces the time-cost remarkably to 969 ms by partitioning the latent into small blocks for parallel decoding.

## 6 CONCLUSIONS

In this work, we propose a masked image modeling transformer for deep video compression. Following the proxy task in pretrained language/image model, the transformer is trained to fully exploit the temporal correlation among frames and spatial tokens in a few autoregressive steps. The MIMT differs from the conventional autoregressive (raster scanning order) in two ways: more flexible bidirectional attention and more efficient parallel decoding. Visualization and qualitative metric on standard datasets demonstrate our method outperforms VTM in terms of both PSNR and SSIM. Our work provides new insights into the use or extension of the transformer in video compression.

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

|  | Params | MACs | Encoding time | Decoding time |
|---|---|---|---|---|
| DVC (Lu et al., 2019) | 4.727 MB | 1.721 T | 3,522 ms | 3,296 ms |
| SSF (Agustsson et al., 2020) | 22.991 MB | 1.424 T | 572 ms | 392 ms |
| DCVC (Li et al., 2021) | 6.526 MB | 2.411 T | 17,856 ms | 97,642 ms |
| DMC (Li et al., 2022) | 17.520 MB | 3.261 T | 1,297 ms | 912 ms |
| MIMT (auto-regressive, *full*) | 38.471 MB | 15.407 T | 1,237 ms | 117,476 ms |
| MIMT (auto-regressive, *block*) | 38.471 MB | 3.640 T | 1,007 ms | 969 ms |
| MIMT (checkerboard) | 38.471 MB | 3.365 T | 964 ms | 798 ms |
| MIMT (proposed) | 38.471 MB | 4.417 T | 1,901 ms | 1,372 ms |

Table 5: Model complexity. All models are tested with 1080p video on an NVIDIA P40 GPU.

Hangbo Bao, Li Dong, and Furu Wei. Beit: Bert pre-training of image transformers. *arXiv preprint arXiv:2106.08254*, 2021.

Gisle Bjontegaard. Calculation of average psnr differences between rd-curves. *VCEG-M33*, 2001.

Tom Brown, Benjamin Mann, Nick Ryder, Melanie Subbiah, Jared D Kaplan, Prafulla Dhariwal, Arvind Neelakantan, Pranav Shyam, Girish Sastry, Amanda Askell, et al. Language models are few-shot learners. *Advances in neural information processing systems*, 33:1877–1901, 2020.

Huiwen Chang, Han Zhang, Lu Jiang, Ce Liu, and William T Freeman. Maskgit: Masked generative image transformer. In *Proceedings of the IEEE/CVF Conference on Computer Vision and Pattern Recognition*, pp. 11315–11325, 2022.

Zhengxue Cheng, Heming Sun, Masaru Takeuchi, and Jiro Katto. Learned image compression with discretized gaussian mixture likelihoods and attention modules. In *Proceedings of the IEEE/CVF Conference on Computer Vision and Pattern Recognition*, pp. 7939–7948, 2020.

Jacob Devlin, Ming-Wei Chang, Kenton Lee, and Kristina Toutanova. Bert: Pre-training of deep bidirectional transformers for language understanding. *arXiv preprint arXiv:1810.04805*, 2018.

Christoph Feichtenhofer, Haoqi Fan, Yanghao Li, and Kaiming He. Masked autoencoders as spatiotemporal learners. *arXiv preprint arXiv:2205.09113*, 2022.

Agrim Gupta, Stephen Tian, Yunzhi Zhang, Jiajun Wu, Roberto Martín-Martín, and Li Fei-Fei. Maskvit: Masked visual pre-training for video prediction. *arXiv preprint arXiv:2206.11894*, 2022.

Dailan He, Yaoyan Zheng, Baochen Sun, Yan Wang, and Hongwei Qin. Checkerboard context model for efficient learned image compression. *2021 IEEE/CVF Conference on Computer Vision and Pattern Recognition (CVPR)*, pp. 14766–14775, 2021.

Kaiming He, Xinlei Chen, Saining Xie, Yanghao Li, Piotr Dollár, and Ross Girshick. Masked autoencoders are scalable vision learners. In *Proceedings of the IEEE/CVF Conference on Computer Vision and Pattern Recognition*, pp. 16000–16009, 2022.

Zhihao Hu, Zhenghao Chen, Dong Xu, Guo Lu, Wanli Ouyang, and Shuhang Gu. Improving deep video compression by resolution-adaptive flow coding. In *European Conference on Computer Vision*, pp. 193–209. Springer, 2020.

Zhihao Hu, Guo Lu, and Dong Xu. Fvc: A new framework towards deep video compression in feature space. In *Proceedings of the IEEE/CVF Conference on Computer Vision and Pattern Recognition*, pp. 1502–1511, 2021.

Zhihao Hu, Guo Lu, Jinyang Guo, Shan Liu, Wei Jiang, and Dong Xu. Coarse-to-fine deep video coding with hyperprior-guided mode prediction. In *Proceedings of the IEEE/CVF Conference on Computer Vision and Pattern Recognition*, pp. 5921–5930, 2022.

Glen G Langdon. An introduction to arithmetic coding. *IBM Journal of Research and Development*, 28(2):135–149, 1984.

Jiahao Li, Bin Li, and Yan Lu. Deep contextual video compression. *Advances in Neural Information Processing Systems*, 34:18114–18125, 2021.

Jiahao Li, Bin Li, and Yan Lu. Hybrid spatial-temporal entropy modelling for neural video compression. *arXiv preprint arXiv:2207.05894*, 2022.

Jianping Lin, Dong Liu, Houqiang Li, and Feng Wu. M-lvc: Multiple frames prediction for learned video compression. In *Proceedings of the IEEE/CVF Conference on Computer Vision and Pattern Recognition*, pp. 3546–3554, 2020.

Ze Liu, Yutong Lin, Yue Cao, Han Hu, Yixuan Wei, Zheng Zhang, Stephen Lin, and Baining Guo. Swin transformer: Hierarchical vision transformer using shifted windows. In *Proceedings of the IEEE/CVF International Conference on Computer Vision*, pp. 10012–10022, 2021.

Guo Lu, Wanli Ouyang, Dong Xu, Xiaoyun Zhang, Chunlei Cai, and Zhiyong Gao. Dvc: An end-to-end deep video compression framework. In *Proceedings of the IEEE/CVF Conference on Computer Vision and Pattern Recognition*, pp. 11006–11015, 2019.

Guo Lu, Xiaoyun Zhang, Wanli Ouyang, Li Chen, Zhiyong Gao, and Dong Xu. An end-to-end learning framework for video compression. *IEEE transactions on pattern analysis and machine intelligence*, 43(10):3292–3308, 2020.

Siwei Ma, Xinfeng Zhang, Chuanmin Jia, Zhenghui Zhao, Shiqi Wang, and Shanshe Wang. Image and video compression with neural networks: A review. *IEEE Transactions on Circuits and Systems for Video Technology*, 30(6):1683–1698, 2019.

Fabian Mentzer, George Toderici, David Minnen, Sung-Jin Hwang, Sergi Caelles, Mario Lucic, and Eirikur Agustsson. Vct: A video compression transformer. *arXiv preprint arXiv:2206.07307*, 2022.

Alexandre Mercat, Marko Viitanen, and Jarno Vanne. Uvg dataset: 50/120fps 4k sequences for video codec analysis and development. In *Proceedings of the 11th ACM Multimedia Systems Conference*, pp. 297–302, 2020.

Xihua Sheng, Jiahao Li, Bin Li, Li Li, Dong Liu, and Yan Lu. Temporal context mining for learned video compression. *arXiv preprint arXiv:2111.13850*, 2021.

Xingjian Shi, Zhourong Chen, Hao Wang, Dit-Yan Yeung, Wai-Kin Wong, and Wang-chun Woo. Convolutional lstm network: A machine learning approach for precipitation nowcasting. *Advances in neural information processing systems*, 28, 2015.

Yibo Shi, Yunying Ge, Jing Wang, and Jue Mao. Alphavc: High-performance and efficient learned video compression. *arXiv preprint arXiv:2207.14678*, 2022.

Zhan Tong, Yibing Song, Jue Wang, and Limin Wang. Videomae: Masked autoencoders are data-efficient learners for self-supervised video pre-training. *arXiv preprint arXiv:2203.12602*, 2022.

Ashish Vaswani, Noam M. Shazeer, Niki Parmar, Jakob Uszkoreit, Llion Jones, Aidan N. Gomez, Lukasz Kaiser, and Illia Polosukhin. Attention is all you need. *ArXiv*, abs/1706.03762, 2017.

Haiqiang Wang, Weihao Gan, Sudeng Hu, Joe Yuchieh Lin, Lina Jin, Longguang Song, Ping Wang, Ioannis Katsavounidis, Anne Aaron, and C-C Jay Kuo. Mcl-jcv: a jnd-based h. 264/avc video quality assessment dataset. In *2016 IEEE international conference on image processing (ICIP)*, pp. 1509–1513. IEEE, 2016.

Zhou Wang, Alan C Bovik, Hamid R Sheikh, and Eero P Simoncelli. Image quality assessment: from error visibility to structural similarity. *IEEE transactions on image processing*, 13(4):600–612, 2004.

Chenfei Wu, Jian Liang, Xiaowei Hu, Zhe Gan, Jianfeng Wang, Lijuan Wang, Zicheng Liu, Yuejian Fang, and Nan Duan. Nuwa-infinity: Autoregressive over autoregressive generation for infinite visual synthesis. *arXiv preprint arXiv:2207.09814*, 2022.

Zhenda Xie, Zheng Zhang, Yue Cao, Yutong Lin, Jianmin Bao, Zhuliang Yao, Qi Dai, and Han Hu. Simmim: A simple framework for masked image modeling. In *Proceedings of the IEEE/CVF Conference on Computer Vision and Pattern Recognition*, pp. 9653–9663, 2022.

Tianfan Xue, Baian Chen, Jiajun Wu, Donglai Wei, and William T Freeman. Video enhancement with task-oriented flow. *International Journal of Computer Vision*, 127(8):1106–1125, 2019.

Ren Yang, Fabian Mentzer, Luc Van Gool, and Radu Timofte. Learning for video compression with hierarchical quality and recurrent enhancement. In *Proceedings of the IEEE/CVF Conference on Computer Vision and Pattern Recognition*, pp. 6628–6637, 2020.

## A  NETWORK

**Image encoder $E$, decoder $D$.** The image encoder and decoder are not the main contributions of our work. We develop our model heavily dependent on the contextual encoding backbone from (Sheng et al., 2021; Li et al., 2022). We use 4 strided convolutional layers for the encoder, resulting in a total factor of $16\times$ downsampling. For the decoder, we use transposed convolutions and add residual blocks at low resolutions. Another critical module is the context network, called the temporal context mining (TCM) module in (Sheng et al., 2021). It generates $1\times$, $2\times$, and $4\times$ downsampled context by warping the feature of the last frame and the predicted optical flow. These contexts are concatenated with the image encoder/decoder at the corresponding scale for temporal reference.

**Transformer Encoder, Decoder.**  We map the input frame of $(256, 256, 3)$ to $(16, 16, 192)$ latent with image encoder. We flatten the 2D grid latent in to sequence $(1, 16 \times 16, 192)$. Thus, the dimension of input $\hat{y}_{t-1}$ is $(1, 256, 192)$. The output of hyper-prior decoder $s_t$ is flatten into $(1, 256, 96)$, and the output of ConvLSTM is $(1, 256, 192)$. We project priors into $d_C = 768$ dimensional with a linear projection layer before feeding to the transformer. These priors are separately processed with a transformer encoder and then concatenated together for temporal mixture with a joint transformer encoder. W-MSA and SW-MSA blocks have another input from the output of the joint encoder. As shown in Table 6, we show the architecture of the transformer encoder and decoder. We use cross attention, i.e., W-MCA and SW-MCA layers, to encourage information mixing of spatial-temporal context and the current frame. The key and value in the cross-attention block are from the output of the joint encoder, where the continuing output of the decoder is the query.

The basic building layer of the transformer is the shifted window attention. The decoder alternate between the W-MSA layer and SW-MSA layer using the output of the joint encoder as the key and value. The window size of W-MSA and SW-MSA is 4. The number of attention heads is 8 for every attention layer. We use two layers of MLP with input size, hidden size, and output size all 768.

**Hyper-prior Encoder, Decoder.** The hyper-prior network compresses and stores a bitstream used as the hierarchical prior $s_t$ (Ballé et al., 2018). The encoder downsamples the latent $y_t$. At the receiver side, the decoder recovers it by upsampling, as shown in Table 7.

**ConvLSTM.** We use a ConvLSTM module (Shi et al., 2015) to aggregate temporal priors of all previous frames before $y_{t-1}$. We employ convolution with one-step LSTM, as described in Table 7.

**Multi-frame training.** We can apply multi-frame (up to 7 frames) and patch-size ($256 \times 256$) for training. Using long video sequences for training can alleviate the accumulated errors propagating throughout inter-frame coding. In the first stage, we use two consecutive frames, including one I frame and one P frame, to train our model for 1 M steps using the hyper-prior entropy model without MIMT. This stage targets a good image coder-decoder that can faithfully reconstruct the image with less emphasis on the bit rate cost. Then we add the MIMT entropy model to make a better bit-rate estimation and use two consecutive frames to minimize the rate-distortion loss for 1 M steps. Finally, we extend the length of the training video sequence to 7 frames for 300 K steps. The learning rate is set as 5e-5. We set the batch size as 8, using the Adam optimizer on a single V100 GPU.

| Encoder for $s_t$ | Encoder for $\hat{y}_{t-1}$ | Encoder for $r_{t-1}$ | Joint Encoder | Decoder |
|---|---|---|---|---|
| out: $(1, 256, 768)$ | out: $(1, 256, 768)$ | out: $(1, 256, 768)$ | out: $(1, 3 \times 256, 768)$ | out: $(1, 256, 2 \times 192)$ |
| MLP | MLP | MLP | MLP | Conv(768, $2 \times 192$, 1, 1, 0) |
| Layer Norm | Layer Norm | Layer Norm | Layer Norm | MLP |
| SW-MSA | SW-MSA | SW-MSA | SW-MSA | Layer Norm |
| Layer Norm | Layer Norm | Layer Norm | Layer Norm | $\rightarrow$ SW-MCA |
| MLP | MLP | MLP | MLP | Layer Norm |
| W-MSA | W-MSA | W-MSA | W-MSA | SW-MSA |
| Layer Norm | Layer Norm | Layer Norm | Layer Norm | Layer Norm |
| Linear(192, 768) | Linear(192, 768) | Linear(192, 768) | | MLP |
| in: $(1, 256, 192)$ | in: $(1, 256, 192)$ | in: $(1, 256, 192)$ | in: $(1, 3 \times 256, 768)$ | Layer Norm |
| | | | | $\rightarrow$ W-MCA |
| | | | | Layer Norm |
| | | | | W-MSA |
| | | | | Layer Norm |
| | | | | Linear(192, 768) |
| | | | | in: $(1, 256, 192)$ |

Table 6: Transformer architecture. Each column is a transformer model, and each row corresponds to a layer in the module. Note that we omit the skip connection in the table. We only display one attention block in the table. A learnable positional embedding is added at the input of each module. In practice, encoder $s_t$, $\hat{y}_{t-1}$, and $r_t$ have 2 blocks; joint encoder has 4 blocks; decoder has 4 blocks.

| Hyper-prior Encoder | Hyper-prior Decoder | ConvLSTM | ResBlock |
|---|---|---|---|
| out: $z_t$ $(1, 96, 4, 4)$ | out: $s_t$ $(1, 192, 16, 16)$ | out: $r_{t-1}$ $(1, 256, 192)$ | out: $(1, 192, 16, 16)$ |
| Conv(96, 96, 5, 2, 2) | ConvTranspose(96, 192, 3, 1, 1, 0) | $h_{t-2} \rightarrow$ LSTM(192, 192, 1) | skip connection |
| ReLU() | ReLU() | reshape | ReLU() |
| Conv(96, 96, 5, 2, 2) | ConvTranspose(96, 96, 5, 2, 2, 1) | ResBlock | Conv(192, 192, 3, 1, 0) |
| ReLU() | ReLU() | ReLU() | ReLU() |
| Conv(192, 96, 3, 1, 1) | ConvTranspose(96, 96, 5, 2, 2, 1) | Conv(192, 192, 3, 1, 1) | Conv(192, 192, 3, 1, 0) |
| in: $y_t$ $(1, 192, 16, 16)$ | in: $\hat{z}_t$ $(1, 96, 4, 4)$ | in: $y_{t-2}$ $(1, 192, 16, 16)$ | in: $(1, 192, 16, 16)$ |

Table 7: Hyper-prior encoder/decoder and ConvLSTM architecture. Conv(input_channels, output_channels, kernel_size, stride, padding). ConvTranspose(input_channels, output_channels, kernel_size, stride, padding, output_padding). LSTM(input_size, hidden_size, num_layers).

## B   RECURRENT PRIOR

Exploiting inter-frame redundancy has always been a critical consideration for video compression. Generally, a deep learning model can use two consecutive frames $\{y_t, y_{t-1}\}$ (Lu et al., 2019; Li et al., 2021), three consecutive frames $\{y_t, y_{t-1}, y_{t-2}\}$ (Mentzer et al., 2022), four consecutive frames $\{y_t, y_{t-1}, y_{t-2}, y_{t-3}\}$ (Hu et al., 2021), or all previous frames using recurrent network (Ma et al., 2019; Yang et al., 2020). We use a one-step ConvLSTM to aggregate all previous frames before $y_{t-1}$. From Table 3, it can be discovered that recurrent prior $r_{t-1}$ bring $11.1\%$ bitrate savings. One alternative to simplify the recurrent prior is to remove the ConvLSTM and use $\hat{y}_{t-2}$ instead. An ablation study is conducted in Table 8. The second row shows that $\hat{y}_{t-2}$ help to save significant bitrate. But there are still $3.9\%$ bitrate saving improvements. This result implies that $\hat{y}_{t-1}, \hat{y}_{t-2}$ contains most temporal context we need to compress $y_t$, but there is still room for improvement if we exploit a longer range of frames. This observation is consistent with the conclusion from FVC (Hu et al., 2021) and RLVC (Yang et al., 2020).

## C   VISUALIZATION OF DECODING

In Fig.7, we visualize the intermediate results of the iterative decoding process. At the $k$-th step, the transformer entropy model predicts the PMFs of undecoded tokens, and we directly fill the non-decoded tokens with the predicted mean $\mu_t$ of PMFs. The quality of the intermediate image can intuitively reflect the entropy prediction ability of the MIMT model.

In the first column, zero tokens of $y_t$ are decoded, i.e., 0 bpp for $y_t$. The bit cost on $v_t$ and $z_t$ is 0.006 and 0.007, respectively. MIMT can predict a coarse estimation of the $x_t$ from encoded

| Inputs for Entropy Model | UVG | MCL-JCV | HEVC-B | Average |
|---|---|---|---|---|
| $s_t$, $\hat{y}_{t-1}$ | +8.6% | +11.3% | +13.5% | +11.1% |
| $s_t$, $\hat{y}_{t-1}$, $\hat{y}_{t-2}$ | +2.2% | +4.1% | +5.4% | +3.9% |
| $s_t$, $\hat{y}_{t-1}$, $r_{t-1}$ | 0% | 0% | 0% | 0% |

Table 8: Ablations on the recurrent prior $r_{t-1}$ of MIMT.

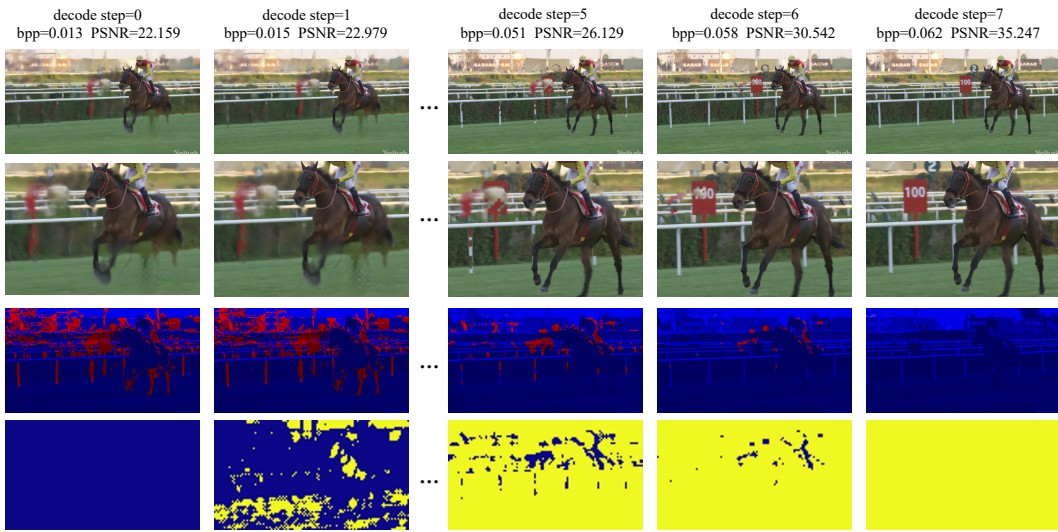

Figure 7: Illustration of decoding process on Jockey in UVG dataset. The first row is the reconstructed images at different steps. The second row shows zoom-in regions in the reconstructed images. The third row plots the pixel-to-pixel loss with respect to the ground-truth image. Brighter red represents a higher loss. The fourth row shows the decoded tokens position, where at the beginning no tokens are decoded (blue) and at last, all tokens are decoded (yellow).

priors $\hat{y}_{t-1}, \hat{z}_t, r_{t-1}$. Although the image is blurred without details, we can be aware of the rough background, the main instances of image, etc. The follow-up images show that as we decode more tokens, the model gets more context, and the current frame is refined progressively.

We can find that the model decodes images following an easy-to-hard process. Considering the high temporal redundancy of frame sequences, a large portion of pixels, mostly background, is easy to predict. The model starts from the tokens with the smallest entropy (e.g., sky and ground) to get context information. In the beginning, we can take large steps because these tokens consume a little bitrate. Gradually, when it "knows" more about the image, it refines the large-entropy tokens (e.g., the fast-moving billboard and horseshoe) which are hard to predict. We should take a small step to make more accurate estimations. This is analogous to the behaviors of a painter who starts with a sketch and then progressively refines it by filling in or tweaking the details. This dynamic autoregressive strategy differs from the raster scanning order might encounter a hard-to-easy dilemma and result in more bit-stream consumption.

## D  DECODING EFFECIENCY

We obtain the opensource code of DVC [1], SSF [2], DCVC [3], and DMC [4] for decoding efficiency comparison.

---

[1] https://github.com/GuoLusjtu/DVC

[2] https://github.com/InterDigitalInc/CompressAI

[3] https://github.com/microsoft/DCVC/tree/main/NeurIPS2021

[4] https://github.com/microsoft/DCVC/tree/main/ACMMM2022

# E  FULL-SIZED RATE-DISTORTION PLOTS

In the following, we show full-size versions of Fig. 5 for better readability.

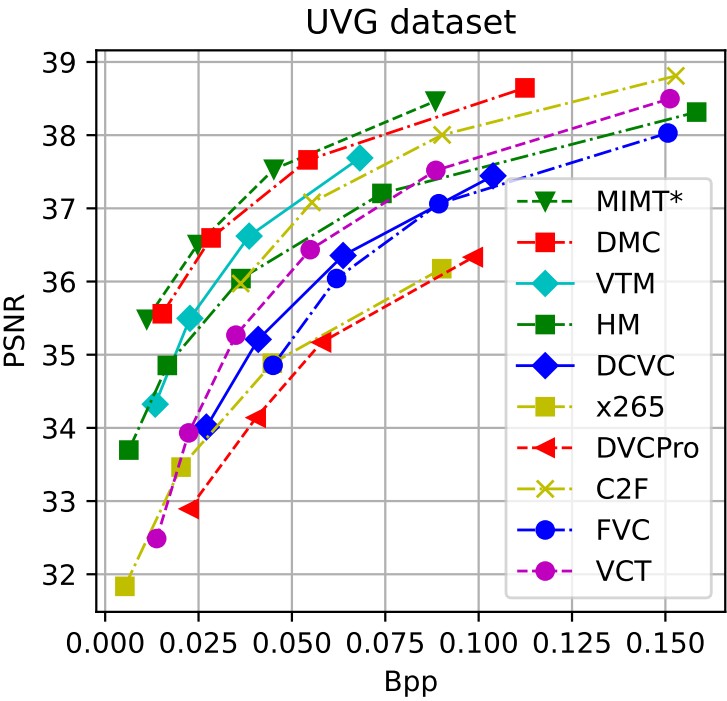

Figure 8: RD performance on UVG dataset in terms of PSNR.

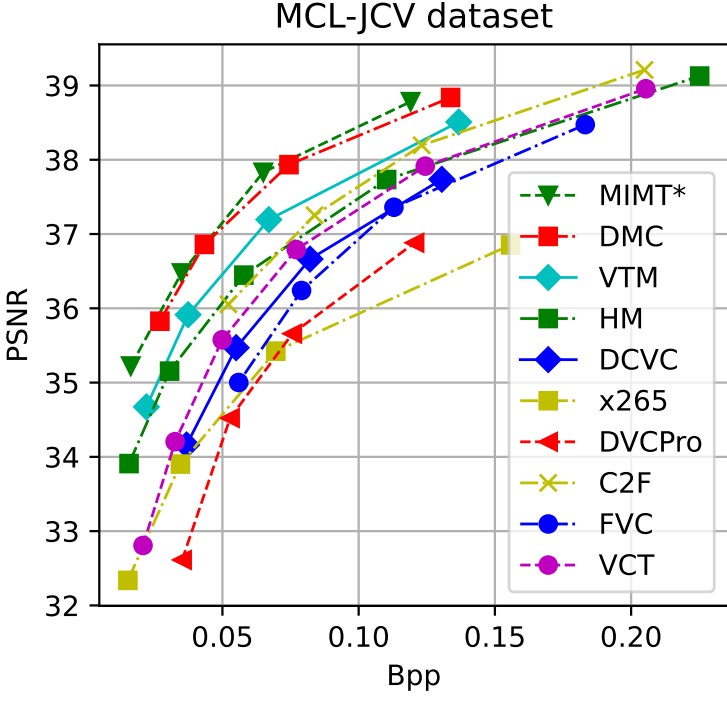

Figure 9: RD performance on MCL-JCV dataset in terms of PSNR.

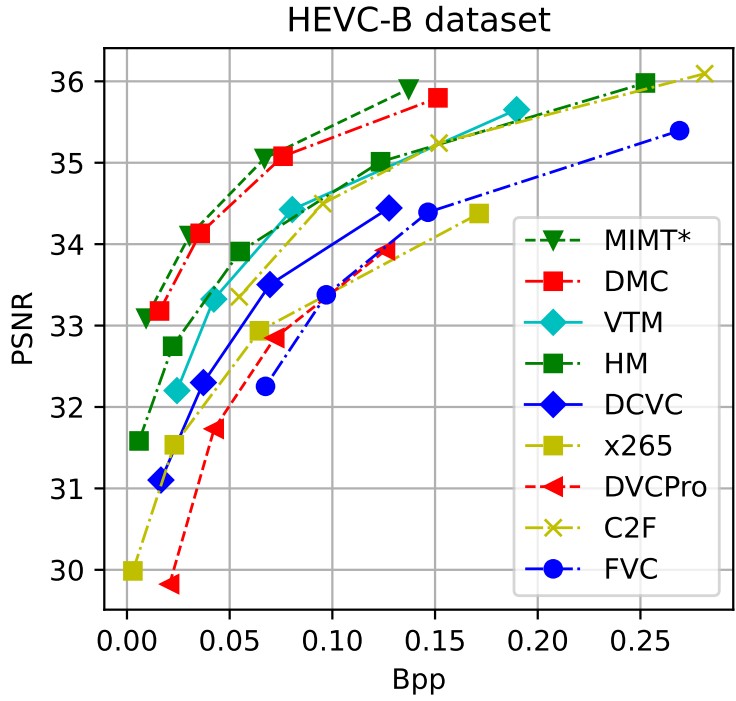

Figure 10: RD performance on HEVC-B dataset in terms of PSNR.

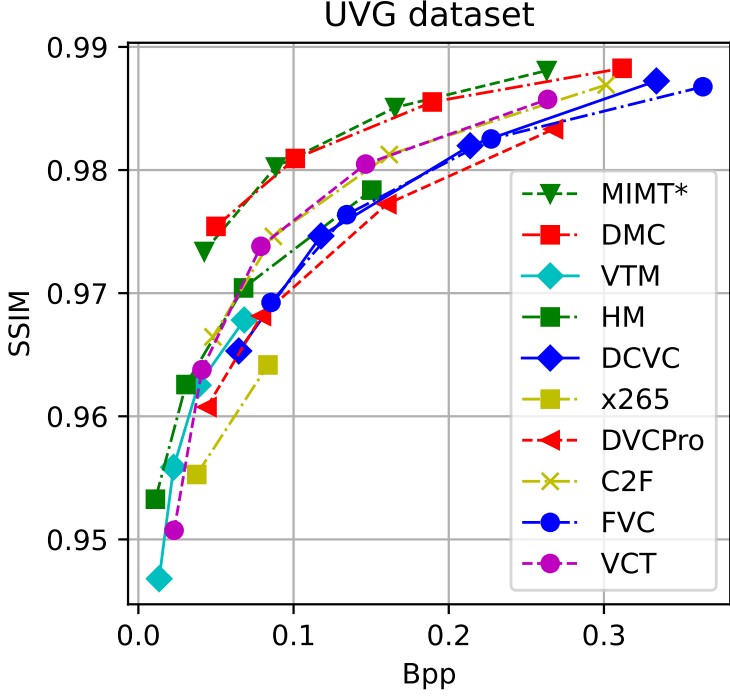

Figure 11: RD performance on UVG dataset in terms of SSIM.

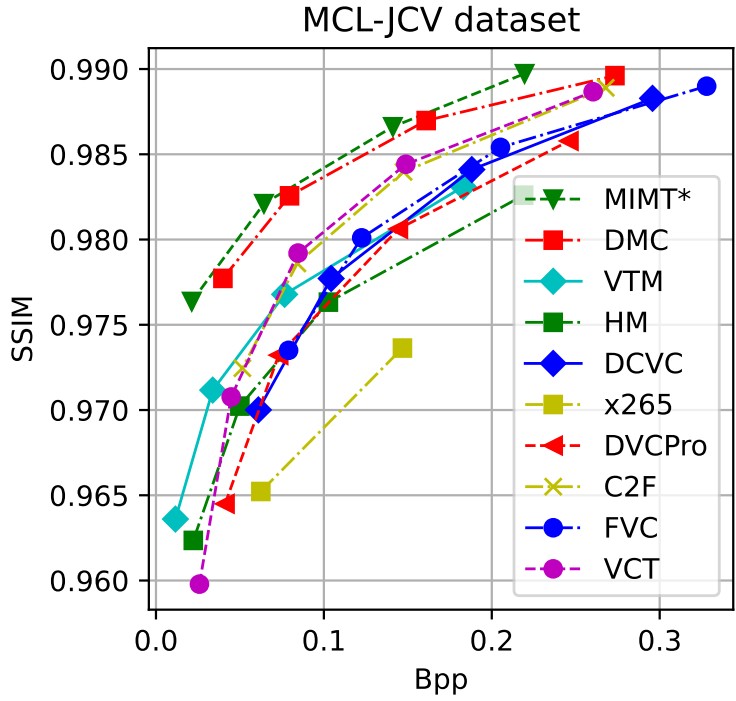

Figure 12: RD performance on MCL-JCV dataset in terms of SSIM.

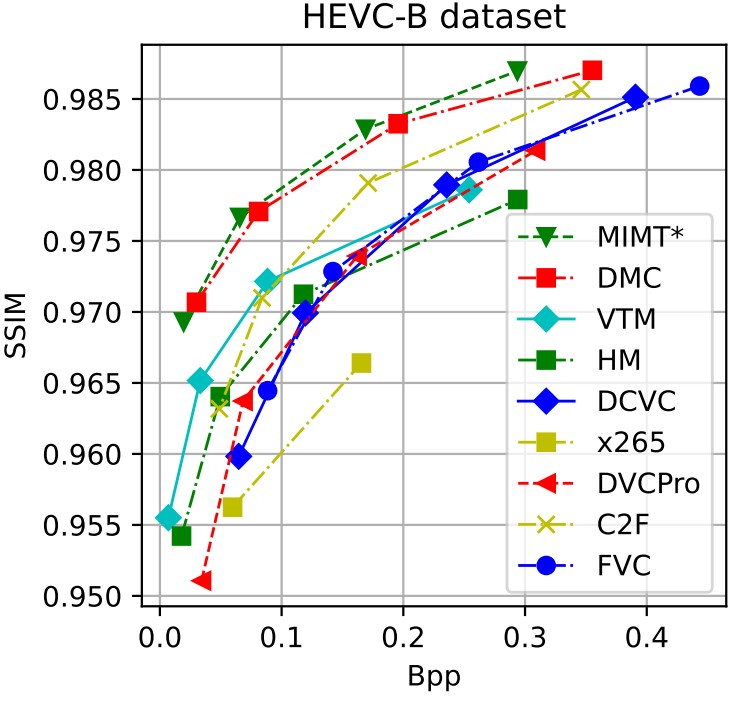

Figure 13: RD performance on HEVC-B dataset in terms of SSIM.

