# OpenReview forum: "MIMT: Masked Image Modeling Transformer for Video Compression"
_ICLR.cc/2023/Conference — ICLR 2023 notable top 25%_

### Official Review · Reviewer_DWzw · 2022-10-16

**Confidence:** 4
**Correctness:** 3
**Technical Novelty And Significance:** 3
**Empirical Novelty And Significance:** 3
**Recommendation:** 6

**Clarity, Quality, Novelty And Reproducibility:**

The paper is easy to read, and from what I can tell the idea of using masked transformers for entropy coding is novel.

**Strength And Weaknesses:**

Strengths:
- The authors have relatively comprehensive evaluations over UVG/MCL/HEVC-B and show that their work is state of the art.
- The authors provide runtime comparisons, which I appreciate
- The intuition of decoding the image in an “easy-to-hard” process is interesting, I appreciate the diagram shown in Fig. 6.


Weaknesses:
- I think the iterative decoding scheduler is interesting but I’d like more theoretical and/or empirical analysis as to why it makes sense to 1) go from low entropy to high-entropy, and 2) using a sine function. Did the authors try out other functions / decoding orders? Why was this function and ordering the one that the authors decided to settle on?
- The recurrent prior + Optical flow predictor isn’t really explained. It doesn’t seem to be a main contribution (the main contribution seems to be MIMT), but it also seems to provide a fairly significant ~10% bitrate savings to all datasets. And this goes into a broader framing of the paper itself. Are the authors proposing the entire e2e system as the novel idea? Or the entropy model as the novel idea? To me, the entropy model seems to be the main idea, and I would love to see an experimental setup that helps me understand what the benefits of that are.
- I don’t entirely get the purpose of section 5.4 (Ablation Studies), since none of these items are central to the author’s core contributions, which is using MIMT as the entropy model. The paper is also missing ablation studies that would help validate the claims that the authors make regarding MIMT. For instance, the benefits of decoding token by token instead of decoding the current frame all in one step. Or the benefits of going low entropy to high entropy vs. say a raster ordering. Or the decision to use a sine function.
- I don’t think this is never explicitly stated in the paper but conceptually you would need to use the decoding scheduler when first encoding each frame into tokens, as well as during decoding time right? Because you would need to determine the order in which to encode the tokens for each frame in order to decode the tokens using the same sequence. It’s not clear from the paper itself, and I was initially confused as to how the decoding scheduler would work.


**Summary Of The Paper:**

The paper proposes MIMT which is a transformer-based model that can predict the PMFs of tokens in arbitrary order, not necessarily in raster-scanning order (e.g. line-by-line). After training the MIMT model, they propose an “iterative decoding scheduler” which decodes tokens from the current frame from smallest entropy to largest.

**Summary Of The Review:**

In summary, I think MIMT is a promising direction and a cool idea - I like the intuition of decoding tokens in non-raster order. But it needs more fleshing out from an experimental and potentially a theoretical perspective. The authors provide relative comprehensive experimental results on overall bitrate/distortion curves, but I would want to see more results that highlight the benefit of MIMT itself. It is also unclear to me how much stuff like the hyperprior, last frame, recurrent prior (none of which are novel ideas) are contributing to the boost of results vs. this entropy model. And a bonus would be to see a larger mathematical justification.

---

> ### Author Response · Authors · 2022-11-16
> **Response to Reviewer DWzw [1/4]**
>
> We would like to thank reviewer **DWzw** for thoughtful comments and helpful feedback on our work. We appreciate that reviewer **DWzw** finds the runtime comparisons, and the intuition of decoding the image in an “easy-to-hard” process in Fig. 6 interesting.
>
>
> > Q1. I think the iterative decoding scheduler is interesting but I’d like more theoretical and/or empirical analysis as to why it makes sense to 1) go from low entropy to high-entropy, and 2) using a sine function. Did the authors try out other functions / decoding orders? Why was this function and ordering the one that the authors decided to settle on?
>
>
> Thank you for your constructive suggestion. In the revised paper, we do more investigations on the masked scheduler for MIMT encoding/decoding.
>
> *In Section 5.2:*
>
> The masked scheduler is a non-trivial design for encoding/decoding. The default scheduler is a *sine* function. We further explore two alternatives, i.e., *linear*, and *exponential* functions. The *sine*  is concave, whereas the *exponential* is convex. A *sine* scheduler is in line with such intuitions: due to the temporal redundancy, the current frame has high similarities with previous frames.  In the beginning, we can take a large step with high certainty. In the last few steps, we have to cope with pixels of large entropy. Thus, we should take small steps to make more precise estimations. We adjust the number of steps from 6 to 12. As shown in Fig. 6, the results demonstrate that the  *sine* performs consistently better than the others. The *linear* function provides decent results but is inferior to the *sine*. The *exponential* performs poorly.  Increasing the number of steps can improve the performance, but for steps larger than 8, the gains are marginal and it will introduce computational overhead for training and inference.
>
> *In the Appendix, Section C, Fig. 7:*
>
> Further visualizations are provided in the Appendix, Section C, Fig. 7. The reason we go from low entropy to high entropy  is to enforce the model to decode images following an easy-to-hard process. Considering the high temporal redundancy of frame sequences, a large portion of pixels, mostly background, is easy to predict. The model starts from the tokens with the smallest entropy (e.g., sky and ground) to get context information. In the beginning,  we can take large steps because these tokens consume a little bitrate.  Gradually, when it "knows" more about the image, it refines the large-entropy tokens (e.g., the fast-moving billboard and horseshoe) which are hard to predict.   We should take a small step to make more accurate estimations. This is analogous to the behaviors of a painter who starts with a sketch and then progressively refines it by filling in or tweaking the details.  This dynamic autoregressive strategy differs from the raster scanning order which might encounter a hard-to-easy dilemma and result in more bit-stream consumption.

---

> > ### Author Response · Authors · 2022-11-16
> > **Response to Reviewer DWzw [2/4]**
> >
> >
> > >  Q2. The recurrent prior + Optical flow predictor isn’t really explained. It doesn’t seem to be a main contribution (the main contribution seems to be MIMT), but it also seems to provide a fairly significant ~10% bitrate savings to all datasets. And this goes into a broader framing of the paper itself. Are the authors proposing the entire e2e system as the novel idea? Or the entropy model as the novel idea? To me, the entropy model seems to be the main idea, and I would love to see an experimental setup that helps me understand what the benefits of that are.
> >
> >
> > Thank you for your feedback. The core idea of our study is a powerful entropy model for spatial-temporal context modeling. The MIMT  is underpinned by a masked transformer encoder-decoder network, and it outperforms the auto-regressive ($+19.5\\%$) or checkerboard ($+17.3\\%$) entropy model by a large margin. Also, the computation complexity is greatly reduced compared with the raster-scan order auto-regressive scheme.
> >
> >
> > We learn from the development of standard and learned codecs that an advanced video compression framework should involve elaborated designs from different perspectives. For example, DMC (Li et al. 2022) employs multiple techniques including contextual encoder/decoder, learnable quantization, spatial and channel-wise autoregressive, etc. We build our framework centric around MIMT with off-the-shell techniques for transformation coding and temporal context aggregation. Admittedly, it would be difficult for MIMT to work alone to achieve SOTA. In the revised paper, we conduct a series of ablation studies to verify the effectiveness of the components.
> >
> > In summary, we conduct the following ablation studies in Section 5.2:
> >
> > 1. Ablation on entropy model. To validate the benefits of  MIMT, we make comparisons with auto-regressive (*full*) that decodes tokens one-by-one, auto-regressive (*block*) that partitions tokens into small blocks, and parallel-friendly checkboard model that decodes the first half of tokens and then sequentially decodes the rest tokens. MIMT is the best-performed entropy model.
> > 2. Ablation on the contextual encoder/decoder.  Video compression methods heavily rely on flow-based prediction to reduce temporal redundancy in video sequences. To get temporal-dependent tokens for MIMT, we follow the good practice to use a flow-based contextual encoder/decoder, which is an important component.
> > 5. Ablation on the manifold input of the entropy model. To fully exploit spatial-temporal context, we take advantage of three kinds of context for MIMT entropy coding, i.e., hierarchical prior $\boldsymbol\{s\}_t$, temporal prior from the last frame $\hat\{\boldsymbol\{y\}\}_\{t-1\}$, and temporal prior from the recurrent  $\boldsymbol\{r\}_t$.

---

> > > ### Author Response · Authors · 2022-11-16
> > > **Response to Reviewer DWzw [3/4]**
> > >
> > >
> > > > Q3. I don’t entirely get the purpose of section 5.4 (Ablation Studies), **since none of these items are central to the author’s core contributions, which is using MIMT as the entropy model.** The paper is also missing ablation studies that would help validate the claims that the authors make regarding MIMT. For instance, the benefits of decoding token by token instead of decoding the current frame all in one step. Or the benefits of going low entropy to high entropy vs. say a raster ordering. Or the decision to use a sine function.
> > >
> > > We appreciate the reviewer’s thoughtful suggestions for improving the organization of our paper. To make more clarifications, we conducted ablation studies on the MIMT.
> > >
> > > The  MIMT uses a dynamic strategy to determine the encoding/decoding order for auto-regressive. Other strategies would be a raster scan order autoregressive entropy model like VCT （Mentzer et al, 2022), or a parallel-friendly checkerboard entropy model (He et al, 2021). For fair comparisons, both auto-regressive and checkerboard models use the same transformer encoder-decoder network, but they train and predict differently. For the auto-regressive model, we train the transformer with a self-attention mask to ensure causality similar to (Vaswani et al. 2017), and then we predict each token one by one in raster scan order. We make two variants of auto-regressive: the *full* variant means operating on all $16\times 16$ tokens, and the *block* variant means splitting the tokens into $4\times 4$  non-overlapping blocks similar to VCT. No windowed attention mask is applied for autoregressive (*block*). As for the checkerboard, we predict the first half tokens with all input tokens masked, and then use the decoded tokens to predict the rest.
> > >
> > > In Table 3, we calculate the BD-rate increase after ablations. In the first row, the MIMT entropy performs better than the *full* auto-regressive with $+6.4\\%$ bitrate savings.  The time cost of auto-regressive (*full*) decoding is prohibitive for practical usage. As an alternative, auto-regressive (*block*) partitions the tokens into sub-blocks and decodes them in parallel. Although it can speed up greatly, it builds on the assumption that all blocks are independent, which is not necessarily true as concluded by VCT (Mentzer et al. 2022).  In this way, the performance of autoregressive (*block*) degrades obviously with $+19.5\\%$ bitrate increase. The parallel checkerboard entropy model is slightly better but still has a $+17.3\\%$ increase. This is intuitively reasonable because the checkerboard model could be considered a simplified case of MIMT with fixed encoding/decoding order.
> > >
> > >
> > >
> > > | Modules | Ablation Options          | **UVG**   | **MCL-JCV** | **HEVC-B** | **Average** |
> > > | ------- | ------------------------- | --------- | ----------- | ---------- | ----------- |
> > > | MIMT    | auto-regressive (*full*)  | $+5.2\\%$  | $+6.9\\%$    | $+7.1\\%$   | $+6.4\\%$    |
> > > | MIMT    | auto-regressive (*block*) | $+16.6\\%$ | $+19.3\\%$   | $+22.8\\%$  | $+19.5\\%$   |
> > > | MIMT    | checkerboard              | $+15.1\\%$ | $+17.7\\%$   | $+19.2\\%$  | $+17.3\\%$   |
> > >
> > >
> > >
> > > In Section 5.3, we also evaluate the model complexity of these entropy models.
> > >
> > > The proposed MIMT equipped with shifted window attention and non-sequential decoding strategy is comparable with DMC, in terms of model complexity and encoding and decoding time. For the other three variants of MIMT, the number of parameters is the same but they are trained differently, thus leading to varying MACs requirements. Not surprisingly, the parallel-friendly checkerboard is the most efficient consuming only 798 ms for decoding. The *full* autoregressive model is prohibitive slow at decoding, and the *block* auto-regressive reduces the time cost remarkably to 969 ms by partitioning the latent into small blocks for parallel decoding.
> > >
> > > |                                 | Pramas   | MACs     | Encoding time | Decoding time |
> > > | ------------------------------- | -------- | -------- | ------------- | ------------- |
> > > | MIMT (auto-regressive, *full*)  | 38.471 M | 15.407 T | 1,237 ms      | 117,471   ms    |
> > > | MIMT (auto-regressive, *block*) | 38.471 M | 3.640 T  | 1,007ms       | 969 ms        |
> > > | checkerboard                    | 38.471 M | 3.365 T  | 964 ms        | 798 ms        |
> > > | MIMT (proposed)                 | 38.471 M | 4.417 T  | 1,901 ms      | 1,372 ms      |

---

> > > > ### Author Response · Authors · 2022-11-16
> > > > **Response to Reviewer DWzw [4/4]**
> > > >
> > > >
> > > > > Q4. I don’t think this is never explicitly stated in the paper but conceptually you would need to use the decoding scheduler when first encoding each frame into tokens, as well as during decoding time right? Because you would need to determine the order in which to encode the tokens for each frame in order to decode the tokens using the same sequence. It’s not clear from the paper itself, and I was initially confused as to how the decoding scheduler would work.
> > > >
> > > > Thank you for raising this point.  Yes, you get it. As with all other codecs, the encoding always will go through the decoding loop. In concrete, the encoding and decoding process uses the same scheduler  $\gamma(i)=\sin({i\cdot \frac{\pi}{2}})$, where $i \in\left\\{0, \frac{1}{n}, \ldots, \frac{n-1}{n}, 1\right\\}$. At the beginning of encoding, we use all tokens as [M], and then we get the first group of tokens' PMFs according to the scheduler.  We use the PMFs to encode the quantized tokens using arithmetic encoding. In the next iteration, we use the quantized tokens as input to predict the next group of tokens and then again encode them into bitstreams. As we can see, the encoding process is the same as decoding as described in Section 3.4, except for using the arithmetic decoder to read bitstream for decoding.
> > > >
> > > >
> > > > > Q5. It is also unclear to me how much stuff like the hyperprior, last frame, recurrent prior (none of which are novel ideas) are contributing to the boost of results vs. this entropy model. And a bonus would be to see a larger mathematical justification.
> > > >
> > > > As shown in Table 4, the hyper-prior $\boldsymbol{s}_t$, the last frame $\hat{\boldsymbol{y}}_\{t-1\}$, and the recurrent prior $\boldsymbol{r}_\{t-1\}$ contribute to  more than $10\\%$. They are used for the entropy model collaboratively because they are complementary by providing spatial-temporal context from different perspectives. In specific, $\boldsymbol{s}_t$ is a spatial hierarchical prior; $\hat{\boldsymbol{y}}_\{t-1\}$ is the most correlated frame in the past; $\boldsymbol{r}_\{t-1\}$ is the aggregated temporal context of all frames before $\boldsymbol{y}_\{t-1\}$.
> > > >
> > > > We strongly agree with you that a larger mathematical justification would help us understand the theoretical benefits of masked image modeling. In our study, we provided empirical insights into applying transformers for spatial-temporal entropy modeling.

---

> ### Author Response · Authors · 2022-11-19
> **Response to Reviewer DWzw**
>
> We sincerely appreciate your thoughtful and objective review.
>
> As suggested, we conducted a series of new experiments to demonstrate the effectiveness of the masked transformer entropy model, as well as the contextual coding and manifold input for the entropy model. The decoding scheme is also a non-trivial design about which we give more details.
>
> We hope our response can adequately address your concerns.
>
> Best regards,
>
> Authors

---

> > ### Comment · Reviewer_DWzw · 2022-11-20
> > **Response to authors**
> >
> > Thanks for the response. The ablation studies in the paper help to clarify MIMT's advantages more (esp when compared to autoregressive model). A theoretical justification would make the paper stronger, but given the performance improvements + the fact that runtime is good I'll raise the score to a 6.

---

> > > ### Author Response · Authors · 2022-11-21
> > > **Response to Reviewer DWzw**
> > >
> > > Thank you for your positive feedback. Your suggestion on theoretical justification actually points out a very interesting research direction in this field.

---

### Official Review · Reviewer_1F8U · 2022-10-24

**Confidence:** 5
**Correctness:** 3
**Technical Novelty And Significance:** 3
**Empirical Novelty And Significance:** 2
**Recommendation:** 8

**Clarity, Quality, Novelty And Reproducibility:**

The paper is easy to follow and is a serious work. MIMT is new and novel. I believe the results should be reproducible.

**Strength And Weaknesses:**

Strengths:
(1) This work introduces a transformer-based, bi-directional entropy model with a masked and incremental coding order. This idea is interesting and novel.
(2) The gain presented looks very promising.

Weaknesses:
(1) In my opinion, MIMT presents a different approach to the group-based auto-regressive model, e.g. Checkerboard Context Model for Efficient Learned Image Compression, CVPR 2021. Similar to the group-based model, MIMT encodes the tokens in an incremental manner. However, the tokens to be encoded in each iteration are identified based on their predicted entropy rates, instead of using a fixed pattern as with the group-based auto-regressive model. The dynamic determination of tokens to be encoded comes at the cost of additional compute for sorting. Moreover, the number of tokens to be coded in each iteration crucially determines the final coding performance. Currently, this is done in a heuristic manner by choosing the scheduling scheme r(i) as sin (i * pi/2). The choice of this scheduling scheme needs justification. I believe a different choice of r(i) would lead to different coding performance. Also, the gain of this dynamic approach over the fixed-pattern approach, e.g. checkerboard context model, needs to be clarified.
(2) The impact of the number of iterations, e.g. the value of n, together with the scheduling scheme r(i) on the final coding result should be studied in an ablation experiment. Moreover, this study should look into how these design choices may affect the complexity in terms of MAC and encoding/decoding runtimes on both CPU and GPU.
(3) The entropy model proposed in [1] uses TWO previously reconstructed latents as inputs to estimate the probability distributions of the latents in the current frame. The proposed method replaces one of the previously reconstructed latents with the Recurrent Prior. It is unclear how much additional gain the Recurrent Prior can bring. Note that the Recurrent Prior may cause error propagation in the error-prone transmission.
(4) While the proposed scheme shows very promising coding gain, its MAC (4.4T) is very high. It also inherits the disadvantage of DMC of high memory requirements due to the need to buffer contextual information. I wonder how the coding result would look like if contextual encoding and decoding is disabled. Note that without contextual encoding, the proposed method would reduce to a scheme similar to [1], which will not suffer from temporal cascading errors.

 [1] Mentzer et. al.  “Vct: A video compression transformer,“ arXiv 2206.07307, 2022.


**Summary Of The Paper:**

This paper extends the notion of conditional coding in DMC and Sheng et al., 2021 by additionally introducing a masked image transformer-based entropy model (MIMT) for learned video compression. The MIMT appears to be inspired by VCT and improves on VCT by a scheduling-based transmission. The gain of the proposed method looks very promising.

**Summary Of The Review:**

This work combines the conditional coding scheme in DMC and an extended version of VCT. The MIMT is inspired by VCT and is novel. While the idea of MIMT is interesting, how it compares with the group-based auto-regressive model needs further clarification. Also, more ablation experiments are needed to understand the complexity-performance trade-off of MIMT. The proposed scheme shows good coding results at the cost of high MAC. It is to be noted that DMC (the existing work) contribute partly (likely considerably) to the overall gain presented in this paper. It is unclear how the coding results may look like if contextual encoding and decoding is disabled and how such a stripped scheme compares to VCT.

---

> ### Author Response · Authors · 2022-11-16
> **Response to Reviewer 1F8U [1/4]**
>
>
> > Q1. In my opinion, MIMT presents a different approach to the group-based auto-regressive model, e.g. Checkerboard Context Model for Efficient Learned Image Compression, CVPR 2021. Similar to the group-based model, MIMT encodes the tokens in an incremental manner. However, the tokens to be encoded in each iteration are identified based on their predicted entropy rates, instead of using a fixed pattern as with the group-based auto-regressive model. The dynamic determination of tokens to be encoded comes at the cost of additional compute for sorting. Also, the gain of this dynamic approach over the fixed-pattern approach, e.g. checkerboard context model, needs to be clarified.
>
> We appreciate the reviewer’s constructive suggestions. We strongly agree with you that we should do more investigations on the entropy model. In Section 5.2, we added more analysis for the entropy model.
>
> *In Section 5.2:*
>
> The  MIMT uses a dynamic strategy to determine the encoding/decoding order for auto-regressive. Other strategies would be a raster scan order autoregressive entropy model like VCT （Mentzer et al, 2022), or a parallel-friendly checkerboard entropy model (He et al, 2021). For fair comparisons, both auto-regressive and checkerboard models use the same transformer encoder-decoder network, but they train and predict differently. For the auto-regressive model, we train the transformer with a self-attention mask to ensure causality similar to (Vaswani et al. 2017), and then we predict each token one by one in raster scan order. We make two variants of auto-regressive: the *full* variant means operating on all $16\times 16$ tokens, and the *block* variant means splitting the tokens into $4\times 4$  non-overlapping blocks similar to VCT.  No windowed attention mask is applied for autoregressive (*block*).  As for the checkerboard, we predict the first half tokens with all input tokens masked, and then use the decoded tokens to predict the rest.
>
> In Table 3, we calculate the BD-rate increase after ablations. In the first row, the MIMT entropy performs better than the *full* auto-regressive with $+6.4\\%$ bitrate savings.  The time cost of auto-regressive (*full*) decoding is prohibitive for practical usage. As an alternative, auto-regressive (*block*) partitions the tokens into sub-blocks and decodes them in parallel. Although it can speed up greatly, it builds on the assumption that all blocks are independent, which is not necessarily true as concluded by VCT (Mentzer et al. 2022).  In this way, the performance of autoregressive (*block*) degrades obviously with $+19.5\\%$ bitrate increase. The parallel checkerboard entropy model is slightly better but still has a $+17.3\\%$ increase. This is intuitively reasonable because the checkerboard model could be considered a simplified case of MIMT with fixed encoding/decoding order.
>
>
> *Table 3. Ablations on the entropy model and the contextual encoder/decoder. The first column is the original design and the second column is the optional modules. The last four columns show the BD rate increase over the anchor of the complete MIMT model.*
>
> | Modules | Ablation Options          | **UVG**   | **MCL-JCV** | **HEVC-B** | **Average** |
> | ------- | ------------------------- | --------- | ----------- | ---------- | ----------- |
> | MIMT    | auto-regressive (*full*)  | $+5.2\\%$  | $+6.9\\%$    | $+7.1\\%$   | $+6.4\\%$    |
> | MIMT    | auto-regressive (*block*) | $+16.6\\%$ | $+19.3\\%$   | $+22.8\\%$  | $+19.5\\%$   |
> | MIMT    | checkerboard              | $+15.1\\%$ | $+17.7\\%$   | $+19.2\\%$  | $+17.3\\%$   |
>
>
> In Section 5.3, we also evaluate the model complexity of these entropy models.
>
> The proposed MIMT equipped with shifted window attention and non-sequential decoding strategy is comparable with DMC, in terms of model complexity and encoding and decoding time. For the other three variants of MIMT, the number of parameters is the same but they are trained differently, thus leading to varying MACs requirements. Not surprisingly, the parallel-friendly checkerboard is the most efficient consuming only 798 ms for decoding. The *full* autoregressive model is prohibitive slow at decoding, and the *block* auto-regressive reduces the time cost remarkably to 969 ms by partitioning the latent into small blocks for parallel decoding.
>
> |                                 | Pramas   | MACs     | Encoding time | Decoding time |
> | ------------------------------- | -------- | -------- | ------------- | ------------- |
> | MIMT (auto-regressive, *full*)  | 38.471 M | 15.407 T | 1,237 ms      | 117,471   ms    |
> | MIMT (auto-regressive, *block*) | 38.471 M | 3.640 T  | 1,007ms       | 969 ms        |
> | checkerboard                    | 38.471 M | 3.365 T  | 964 ms        | 798 ms        |
> | MIMT (proposed)                 | 38.471 M | 4.417 T  | 1,901 ms      | 1,372 ms      |

---

> > ### Author Response · Authors · 2022-11-16
> > **Response to Reviewer 1F8U [2/4]**
> >
> >
> >
> > > Q2. Moreover, the number of tokens to be coded in each iteration crucially determines the final coding performance. Currently, this is done in a heuristic manner by choosing the scheduling scheme r(i) as sin (i * pi/2). The choice of this scheduling scheme needs justification. I believe a different choice of r(i) would lead to different coding performance.
> >
> > Thank you for your review. In the revised paper, we do more investigations on the masked scheduler for MIMT encoding/decoding.
> >
> > *In Section 5.2:*
> >
> > The masked scheduler is a non-trivial design for  encoding/decoding. The default scheduler is a *sine* function. We further explore two alternatives, i.e., *linear*, and *exponential* functions. The *sine*  is concave, whereas the *exponential* is convex. As shown in Fig. 6, the results demonstrate that the  *sine* performs consistently better than the others. The *linear* function provides decent results but is inferior to the *sine*. The *exponential* performs poorly.  A *sine* scheduler is in line with such intuitions: due to the high temporal redundancy, many pixels have slight differences from the previous frames. In the beginning, we can take a large step with high certainty. In the last few steps, we have to cope with pixels of large entropy where we should take small steps to make more precise estimations.
> >
> > Further visualizations are provided in the Appendix, Section C. In Fig. 7, We can find that the model decodes images following an easy-to-hard process. Considering the high temporal redundancy of frame sequences, a large portion of pixels, mostly background, is easy to predict. The model starts from the tokens with the smallest entropy (e.g., sky and ground) to get context information. In the beginning,  we can take large steps because these tokens consume a little bitrate.  Gradually, when it "knows" more about the image, it refines the large-entropy tokens (e.g., the fast-moving billboard and horseshoe) which are hard to predict.   We should take a small step to make more accurate estimations. This is analogous to the behaviors of a painter who starts with a sketch and then progressively refines it by filling in or tweaking the details.  This dynamic autoregressive strategy differs from the raster scanning order which might encounter a hard-to-easy dilemma and result in more bit-stream consumption.
> >
> >
> >
> > > Q3. The impact of the number of iterations, e.g. the value of n, together with the scheduling scheme r(i) on the final coding result should be studied in an ablation experiment. Moreover, this study should look into how these design choices may affect the complexity in terms of MAC and encoding/decoding runtimes on both CPU and GPU.
> >
> >
> > *In Section 5.2:*
> >
> > We adjust the number of steps from 6 to 12. As shown in Fig. 6, increasing the number of steps can improve the performance, but for steps larger than 8, the gains are marginal and it will introduce computational overhead for training and inference. We evaluate the MACs and encoding/decoding runtimes in Section 5.3, as clarified in Q1.

---

> > > ### Author Response · Authors · 2022-11-16
> > > **Response to Reviewer 1F8U [3/4]**
> > >
> > >
> > > > Q4. The entropy model proposed in VCT uses TWO previously reconstructed latents as inputs to estimate the probability distributions of the latents in the current frame. The proposed method replaces one of the previously reconstructed latents with the Recurrent Prior. It is unclear how much additional gain the Recurrent Prior can bring. Note that the Recurrent Prior may cause error propagation in the error-prone transmission.
> > >
> > > Thank you for raising this point.  Exploiting inter-frame redundancy has always been a critical consideration for video compression. Generally, a deep learning model can use two consecutive frames $\{y_{t}, y_{t−1}\}$ (Lu et al., 2019; Li et al., 2021), three consecutive frames $\{y_t, y_{t−1}, y_{t−2}\}$ (Mentzer et al., 2022), four consecutive frames $\{y_t, y_{t−1}, y_{t−2}, y_{t−3}\}$ (Hu et al., 2021), or all previous frames using recurrent network (Ma et al., 2019; Yang et al., 2020). We use a one-step ConvLSTM to aggregate all previous frames before $y_{t−1}$. From Table 3, it can be discovered that recurrent prior $\boldsymbol{r}_{t−1}$ bring 11.1% bitrate savings.
> > >
> > > One alternative to simplify the recurrent prior is to remove the ConvLSTM and use $\hat{y}_\{t−2\}$ instead. An ablation study has been conducted in Table 8. The second row shows that $\hat{y}_\{t−2\}$ helps to save significant bitrate. But there are still $3.9\\%$ bitrate saving improvements. This result implies that $\hat{y}_\{t−1\}, \hat{y\}_\{t−2\}$ contains most temporal context we need to compress $y_t$, but there is still room for improvement if we exploit a longer range of frames.  This observation is consistent with the conclusion from FVC (Hu et al., 2021) and RLVC (Yang et al., 2020).
> > >
> > > Note that we use a multi-frame training strategy (in Appendix, Section A), the joint rate-distortion loss of 7 frames is optimized at the last stage. In this way,  the inter-frame propagation error can be alleviated combined with a long range of temporal prior.
> > >
> > >
> > > *Table 8 Ablations on the recurrent prior $\boldsymbol{r}_{t-1}$ of MIMT.*
> > > | Inputs for Entropy Model                                                   | **UVG**  | MCL-JCV   | **HEVC-B** | Average   |
> > > | -------------------------------------------------------------------------- | -------- | --------- | ---------- | --------- |
> > > | $\boldsymbol{s}_t, \hat{\boldsymbol{y}}_\{t-1\}$                             | $+8.6\\%$ | $+11.3\\%$ | $+13.5\\%$  | $+11.1\\%$ |
> > > | $\boldsymbol{s}_t, \hat{\boldsymbol{y}}_\{t-1\}, \hat{\boldsymbol{y}}_\{t-2\}$ | $+2.2\\%$ | $+4.1\\%$  | $+5.4\\%$    | $+3.9\\%$  |
> > > | $\boldsymbol{s}_t, \hat{\boldsymbol{y}}_\{t-1\}, \boldsymbol{r}_\{t-1\}$       | $0\\%$    | $0\\%$     | $0\\%$      | $0\\%$     |
> > >
> > > [1] Guo Lu, Wanli Ouyang, Dong Xu, Xiaoyun Zhang, Chunlei Cai, and Zhiyong Gao. Dvc: An end-to-end deep video compression framework. In Proceedings of the IEEE/CVF Conference on Computer Vision and Pattern Recognition, pp. 11006–11015, 2019.
> > >
> > > [2] Jiahao Li, Bin Li, and Yan Lu. Deep contextual video compression. Advances in Neural Information Processing Systems, 34:18114–18125, 2021.
> > >
> > > [3] Fabian Mentzer, George Toderici, David Minnen, Sung-Jin Hwang, Sergi Caelles, Mario Lucic, and Eirikur Agustsson. Vct: A video compression transformer. arXiv preprint arXiv:2206.07307, 2022.
> > >
> > > [4] Siwei Ma, Xinfeng Zhang, Chuanmin Jia, Zhenghui Zhao, Shiqi Wang, and Shanshe Wang. Image and video compression with neural networks: A review. IEEE Transactions on Circuits and Systems for Video Technology, 30(6):1683–1698, 2019.
> > >
> > > [5] Ren Yang, Fabian Mentzer, Luc Van Gool, and Radu Timofte. Learning for video compression with hierarchical quality and recurrent enhancement. In Proceedings of the IEEE/CVF Conference on Computer Vision and Pattern Recognition, pp. 6628–6637, 2020

---

> > > > ### Author Response · Authors · 2022-11-16
> > > > **Response to Reviewer 1F8U [4/4]**
> > > >
> > > >
> > > > > Q5. While the proposed scheme shows very promising coding gain, its MAC (4.4T) is very high. It also inherits the disadvantage of DMC of high memory requirements due to the need to buffer contextual information. I wonder how the coding result would look like if contextual encoding and decoding is disabled. Note that without contextual encoding, the proposed method would reduce to a scheme similar to [1], which will not suffer from temporal cascading errors.
> > > >
> > > > Thank you for raising this point. Video compression methods heavily rely on flow-based prediction to reduce temporal redundancy in video sequences. We follow the good practice to use flow-based transformation coding (Sheng et al., 2021; Li et al., 2021; 2022).
> > > >
> > > > In the last row of Table 3, we conduct an ablation study on the image encoder by removing the temporal context.  The results show the contextual encoder provides significant bitrate saving of $+16.9\\%$ over the non-contextual encoder with independent image compression. This demonstrates that contextual coding is complementary to the proposed MIMT to boost the rate-distortion performance.
> > > >
> > > > For reference, the ablation study in DCVC (Li et al. 2021) verifies that contextual coding can bring about $+12.9\\%$ bitrate savings for their model. The results also show, even when the model reduces to a non-contextual encoder/decoder similar to VCT, there is still a large margin of improvement over VCT.
> > > >
> > > >
> > > > *Table.  The last four columns show the BD rate increase over the anchor of the complete MIMT model.*
> > > >
> > > > | Ablation Options                                   | UVG       | MCL-JCV   | HEVC-B    | Average   |
> > > > | -------------------------------------------------- | --------- | --------- | --------- | --------- |
> > > > | non-contextual encoder/decoder                     | $+13.0\\%$ | $+18.4\\%$ | $+19.5\\%$ | $+16.9\\%$ |
> > > > | VCT  |  $+99.1\\%$         | $+75.4\\%$          |    NA     |   $+87.2\\%$        |
> > > >
> > > >
> > > >
> > > > **Reference:**
> > > >
> > > > Xihua Sheng, Jiahao Li, Bin Li, Li Li, Dong Liu, and Yan Lu. Temporal context mining for learned video compression. arXiv preprint arXiv:2111.13850, 2021.
> > > >
> > > > Jiahao Li, Bin Li, and Yan Lu. Deep contextual video compression. Advances in Neural Information Processing Systems, 34:18114–18125, 2021.
> > > >
> > > > Jiahao Li, Bin Li, and Yan Lu. Hybrid spatial-temporal entropy modelling for neural video compression. arXiv preprint arXiv:2207.05894, 2022.10

---

> ### Author Response · Authors · 2022-11-19
> **Dear Reviewer 1F8U**
>
> It's our great pleasure to receive your constructive feedback and very helpful suggestions.
>
> Following your questions, we conduct additional experiments to demonstrate the effectiveness of the proposed MIMT entropy model, in comparison with two other alternatives, i.e., raster scan order auto-regressive and checkboard. Also, we show the contribution of the flow-based contextual encoder in comparison with a non-contextual encoder/decoder. Other detailed responses are also listed.
>
> If there are any further concerns, we will be happy to clarify them without delay.
>
> Best regards,
>
> Authors

---

### Official Review · Reviewer_Gx3J · 2022-10-24

**Confidence:** 3
**Correctness:** 3
**Technical Novelty And Significance:** 3
**Empirical Novelty And Significance:** 3
**Recommendation:** 6

**Clarity, Quality, Novelty And Reproducibility:**

*Clarity:*

I believe that paper could benefit from some more polishing of the text. For example, I would not call masked image modeling a “bi-directional” method as I feel this could be confused with temporally bi-directional modeling in the compression context.

*Quality and novelty:*

The method is state of the art. Masked image modeling is the main novel contribution, the other components have been used previously in the context of neural compression. However, the benefits of masked image modeling are not properly evaluated and quantified (see weaknesses above).

*Reproducibility:*

The most important architecture and training details are documented in the paper. However, given the many modules used, the proposed method will likely be hard to implement. For example what is the number of attention heads and MLP hidden dimension used by the transformer and Swin transformer layers? What are the architectural details of the ConvLSTM? I could only find the shape of ConvLSTM outputs in the paper.


**Strength And Weaknesses:**

*Strengths:*

The proposed method obtains state-of-the-art results, outperforming several recent engineered and neural codecs across a variety of established benchmark data sets and metrics. Moreover, the proposed method seems to be the first one to use masked image modeling in the context of entropy coding for neural compression. This intuitively makes sense as masked image modeling was recently successfully deployed for likelihood-based generative image modeling (Chang et al. 2022) and where it allows for good quality-speed tradeoffs.

*Weaknesses:*

Besides masked image modeling, the method employs a large set of known modules/tricks (optical flow with multi-scale motion, hyper-prior, recurrent prior, windowed self-attention,...) some of which are ablated and shown to be effective. However, the paper’s main technical novelty - masked image modeling - is not ablated. How does this compare to (potentially structured) autoregressive modeling? By performing autoregression e.g. over spatial blocks or channels, similar rate/compute tradeoffs might be attainable similar to those realized by masked image modeling. Indeed, Table 4 shows that some of the modules/tricks lead to BD rate increases of 10-15%, so they might be to a large part be responsible for the superiority of the proposed method.


**Summary Of The Paper:**

The paper proposes a neural video compression method which leverages masked image modeling for entropy coding. The advantage of masked image modeling in the context of image generation, where it was previously used, is that it provides better quality-complexity trade offs than fully autoregressive modeling. When combined with other techniques such as motion compensation, hyper-prior modeling, and a recurrent prior the proposed method achieves state-of-the-art BD-rate performance compared to recent engineered and neural video compression algorithms, on UVG, MCL-JCV, and HEVC-B, in PSNR and SSIM.

**Summary Of The Review:**

The paper proposes to use masked image modeling for entropy coding in the context of neural video compression and presents a corresponding system which outperforms recent state-of-the-art engineered and neural codecs. While this is a strong result, the paper lacks an in-depth evaluation of the masked image modeling component - its main technical contribution.

---

> ### Author Response · Authors · 2022-11-16
> **Response to Reviewer Gx3J [1/2]**
>
>
> We are happy to see that the reviewer **Gx3J** finds our work well-motivated with masked image modeling and we appreciate for pointing out certain issues which we revise and clarify as follows:
>
>
> > Q1.  the paper’s main technical novelty - masked image modeling - is not ablated. How does this compare to (potentially structured) autoregressive modeling? By performing autoregression e.g. over spatial blocks or channels, similar rate/compute tradeoffs might be attainable similar to those realized by masked image modeling.
>
>
> We appreciate the reviewer’s thoughtful suggestions for improving the ablation studies. In Section 5.2, we added a more in-depth analysis of the entropy model.
>
> The  MIMT uses a dynamic strategy to determine the encoding/decoding order for auto-regressive. Other strategies would be a raster scan order autoregressive entropy model like VCT （Mentzer et al, 2022), or a parallel-friendly checkerboard entropy model (He et al, 2021). For fair comparisons, both auto-regressive and checkerboard models use the same transformer encoder-decoder network, but they train and predict differently. For the auto-regressive model, we train the transformer with a self-attention mask to ensure causality similar to (Vaswani et al. 2017), and then we predict each token one by one in a raster scan order. We make two variants of auto-regressive: the *full* variant means operating on all $16\times 16$ tokens, and the *block* variant means splitting the tokens into $4\times 4$  non-overlapping blocks similar to VCT. No windowed attention mask is applied for autoregressive (*block*). As for the checkerboard, we predict the first half tokens with all input tokens masked, and then use the decoded tokens to predict the rest.
>
> In Table 3, we calculate the BD-rate increase after ablations. In the first row, the MIMT entropy performs better than the *full* auto-regressive with $+6.4\\%$ bitrate saving.  The time cost of auto-regressive (*full*) decoding is prohibitive for practical usage. As an alternative, auto-regressive (*block*) partitions the tokens into sub-blocks and decodes them in parallel. Although it can speed up greatly, it builds on the assumption that all blocks are independent, which is not necessarily true as concluded by VCT (Mentzer et al. 2022).  In this way, the performance of autoregressive (*block*) degrades obviously with $+19.5\\%$ bitrate increase. The parallel checkerboard entropy model is slightly better but still has a $+17.3\\%$ increase. This is intuitively reasonable because the checkerboard model could be considered a simplified case of MIMT with fixed encoding/decoding order.
>
>
> *Table 3. Ablations on the entropy model and the contextual encoder/decoder. The first column is the original design and the second column is the optional modules. The last four columns show the BD rate increase over the anchor of the complete MIMT model.*
>
> | Modules | Ablation Options          | **UVG**   | **MCL-JCV** | **HEVC-B** | **Average** |
> | ------- | ------------------------- | --------- | ----------- | ---------- | ----------- |
> | MIMT    | auto-regressive (*full*)  | $+5.2\\%$  | $+6.9\\%$    | $+7.1\\%$   | $+6.4\\%$    |
> | MIMT    | auto-regressive (*block*) | $+16.6\\%$ | $+19.3\\%$   | $+22.8\\%$  | $+19.5\\%$   |
> | MIMT    | checkerboard              | $+15.1\\%$ | $+17.7\\%$   | $+19.2\\%$  | $+17.3\\%$   |
>
>
>
> In Section 5.3, we also evaluate the model complexity of these entropy models.
>
> The proposed MIMT equipped with shifted window attention and non-sequential decoding strategy is comparable with DMC, in terms of model complexity and encoding and decoding time. For the other three variants of MIMT, the number of parameters is the same but they are trained differently, thus leading to varying MACs requirements. Not surprisingly, the parallel-friendly checkerboard is the most efficient consuming only 798 ms for decoding. The *full* autoregressive model is prohibitive slow at decoding, and the *block* auto-regressive reduces the time cost remarkably to 969 ms by partitioning the latent into small blocks for parallel decoding.
>
> |                                 | Pramas   | MACs     | Encoding time | Decoding time |
> | ------------------------------- | -------- | -------- | ------------- | ------------- |
> | MIMT (auto-regressive, *full*)  | 38.471 M | 15.407 T | 1,237 ms      | 117,471   ms    |
> | MIMT (auto-regressive, *block*) | 38.471 M | 3.640 T  | 1,007ms       | 969 ms        |
> | checkerboard                    | 38.471 M | 3.365 T  | 964 ms        | 798 ms        |
> | MIMT (proposed)                 | 38.471 M | 4.417 T  | 1,901 ms      | 1,372 ms      |

---

> > ### Author Response · Authors · 2022-11-16
> > **Response to Reviewer Gx3J [2/2]**
> >
> >
> >
> > > Q2. I believe that paper could benefit from some more polishing of the text. For example, I would not call masked image modeling a “bi-directional” method as I feel this could be confused with temporally bi-directional modeling in the compression context.
> >
> >
> > Thank you for your suggestion.
> >
> >
> > Bidirectional attention is a carry-over term from BERT. Transformer and BERT can directly access all positions in the sequence, equivalent to having full random access memory of the sequence during encoding/decoding. We inherit this bidirectional idea (masked) which is different from classic auto-regressive in video compression.  B-frame (bidirectional predicted frame) is a more specific term in video compression referring to compression with the bidirectional prediction for the current frame. We will clarify our expression in different scenarios.
> >
> > > Q3. The most important architecture and training details are documented in the paper. However, given the many modules used, the proposed method will likely be hard to implement. For example what is the number of attention heads and MLP hidden dimension used by the transformer and Swin transformer layers? What are the architectural details of the ConvLSTM? I could only find the shape of ConvLSTM outputs in the paper.
> >
> > Thank you for raising this point. Some modules we used are off-the-shell from previous studies. To make it self-explained, we added more implementation details in the revised paper.
> >
> > *In the Appendix, Section A:*
> >
> > The window size of W-MSA and SW-MSA is 4.  The number of attention heads is 8 for every attention layer.  We use two layers of MLP with input size,  hidden size, and output size all 768.
> >
> >
> > **Hyper-prior Encoder, Decoder.** The hyper-prior network compresses and stores a bitstream used as the hierarchical prior $\boldsymbol{s}_t$ (Ball ́e et al. 2018). The encoder downsamples the latent $\boldsymbol{y}_t$. At the receiver side,  the decoder recovers it by upsampling.
> >
> >
> > **ConvLSTM**. We use a ConvLSTM module (shi et al, 2015) to aggregate temporal priors of all previous frames  before $\boldsymbol{y}_{t-1}$. We  employ convolution with one-step LSTM.
> >
> > *Table 7. Hyper-prior encoder/decoder and ConvLSTM architecture. Conv(input\_channels, output\_channels, kernel\_size, stride, padding). ConvTranspose(input\_channels, output\_channels, kernel\_size, stride, padding, output\_padding). LSTM(input\_size, hidden\_size, num\_layers).*
> > | Hyper-prior Encoder                     | Hyper-prior Decoder                          | ConvLSTM                                            | ResBlock              |
> > | --------------------------------------- | -------------------------------------------- | --------------------------------------------------- | --------------------- |
> > | out: $\boldsymbol{z}_t$  (1, 96, 4, 4)  | out: $\boldsymbol{s}_t$ (1, 192, 16, 16)     | out: $\boldsymbol{r}_{t-1}$ (1, 256, 192)           | out: (1, 192, 16, 16) |
> > | Conv(96, 96, 5, 2, 2)                   | ConvTranspose(96, 192, 3, 1, 1, 0)           | $\boldsymbol{h}_{t-2}\rightarrow$ LSTM(192, 192, 1) |        skip connection : + in              |
> > | ReLU()                                  | ReLU()                                       | Reshape                                             |        ReLU()               |
> > | Conv(96, 96, 5, 2, 2)                   | ConvTranspose(96, 96, 5, 2, 2, 1)            | ResBlock                                            |        Conv(192, 192, 3, 1, 0)               |
> > | ReLU()                                  | ReLU()                                       | ReLU()                                              |         ReLu()              |
> > | Conv(192, 96, 3, 1, 1)                  | ConvTranspose(96, 96, 5, 2, 2, 1)            | Conv(192, 192, 3, 1,1)                              |          Conv(192, 192, 3, 1, 0)             |
> > | in: $\boldsymbol{y}_t$ (1, 192, 16, 16) | in: $\hat{\boldsymbol{z}}_{t}$ (1, 96, 4, 4) | in: $\boldsymbol{y}_{t-2}$ (1, 192, 16, 16)         | in: (1, 192, 16, 16)  |

---

> > > ### Comment · Reviewer_Gx3J · 2022-11-18
> > > **Response to the author response**
> > >
> > > I thank the authors for their detailed response. Given the comparison with different autoregressive entropy model variants, I am now more convinced that MIMT has practical advantages. I decided to raise my score given the strong rate-distortion performance of the method.
> > >
> > > The authors have not provided a revised version of the paper as far as I can tell. So in case the paper is accepted, I would expect a substantial revision, improving clarity and adding the new details provided in the response.

---

> > > > ### Author Response · Authors · 2022-11-18
> > > > **Dear Reviewer Gx3J**
> > > >
> > > > It's a great pleasure to receive your feedback.
> > > >
> > > > Although we tried to address every concern with care, there could be unexpected missing details in the revised paper, partially because of the short time period of the discussion stage. As suggested, we will further proofread the paper if it is accepted.
> > > >
> > > > Best regards,
> > > >
> > > > Authors

---

### Official Review · Reviewer_33TR · 2022-10-25

**Confidence:** 4
**Correctness:** 4
**Technical Novelty And Significance:** 2
**Empirical Novelty And Significance:** 3
**Recommendation:** 6

**Clarity, Quality, Novelty And Reproducibility:**

- Clarity: Paper is clearly written and easy to understand
- Quality: Paper pushes SOTA on video compression.
- Novelty: Incremental at best, but to effective improvement in compression performance.
- Reproducibility: Training details are missing. Further, model details of how prior information is generated is missing too.


**Strength And Weaknesses:**

Strengths
- The paper is clearly written and easy to understand.
- Improved performance over previous strong baselines.
- A good set of ablation experiments that study some of the method choices and provide insights into them (e.g. visualization and effect of prior information).

Weaknesses
- Novelty: Incremental. Two main contributions are use of SWIN transformer inspired architecture for efficiency and Masked autoencoding based entropy model. Individually both are incremental. However, the improvements resulting from this combination make up for the incremental novelty in my view.
- Missing ablation of decoding schedule: Paper uses a sine based schedule to decide patches to decode. It would be useful to understand how critical this choice is to the method's performance. For example, a simple comparison would be simple linear schedule.
- Training details are missing, leading to poor reproducibility. Model details of how prior information is generated is missing too. Please clarify in appendix at the very least.


**Summary Of The Paper:**

The paper proposes to use an iterative decoding scheme for entropy model based on masked image modeling to improve video compression. This scheme has the advantage that a fixed autoregressive order is not used and thus leading to a better modeling of the latents. Further, SWIN transformer based model is used to improve the efficiency of the method.

**Summary Of The Review:**

Overall, despite the marginal novelty, I am positive about this paper. Existing techniques are used effectively to improve compression performance. I think these improvements are substantial enough to warrant acceptance.

---

> ### Author Response · Authors · 2022-11-16
> **Response to Reviewer 33TR [1/2]**
>
>
> > Q1. Novelty: Incremental. Two main contributions are use of SWIN transformer inspired architecture for efficiency and Masked autoencoding based entropy model. Individually both are incremental. However, the improvements resulting from this combination make up for the incremental novelty in my view.
>
> We thank the reviewer  **33TR** for valuable feedback on our manuscript. We are happy to hear that the reviewer agrees with the compression improvements and that the use of MIMT combing with some existing techniques is substantial enough to warrant acceptance.
>
> Inspired by the success of masked image modeling, e.g., BEIT (Bao et al. 2021), and SimMIM (Xie et al. 2022), we develop a transformer encoder-decoder network for spatial-temporal entropy modeling. The proposed MIMT provides an efficient way of dynamic order decoding, which is significantly different from fixed-order decoding methods (e.g., full autoregressive, block autoregressive, and checkerboard). It could improve both bitrate saving and computational efficiency.
>
> We learn from the development of standard codecs that an advanced video compression framework should involve elaborated designs from different perspectives.  Therefore, we build our framework centric around MIMT with off-the-shell techniques for transformation coding and temporal context aggregation.
>
>
> [1] Hangbo Bao, Li Dong, and Furu Wei. Beit: Bert pre-training of image transformers. arXiv preprint arXiv:2106.08254, 2021
>
> [2] Zhenda Xie, Zheng Zhang, Yue Cao, Yutong Lin, Jianmin Bao, Zhuliang Yao, Qi Dai, and Han Hu. Simmim: A simple framework for masked image modeling. In Proceedings of the IEEE/CVF Conference on Computer Vision and Pattern Recognition, pp. 9653–9663, 2022.
>
> > Q2. Missing ablation of decoding schedule: Paper uses a sine based schedule to decide patches to decode. It would be useful to understand how critical this choice is to the method's performance. For example, a simple comparison would be simple linear schedule.
>
> Thank you for your constructive suggestion.  The masked scheduler is a non-trivial design for encoding/decoding. The default scheduler is a *sine* function with 8 decoding steps. In the revised paper, we do more investigations on the masked scheduler for MIMT encoding/decoding.
>
> We further explore two alternatives, i.e., *linear*, and *exponential* functions. The *sine*  is concave, whereas the *exponential* is convex. We adjust the number of steps from 6 to 12. As shown in Fig. 6, the results demonstrate that the  *sine* performs consistently better than the others. The *linear* function provides decent results but is inferior to the *sine*. The *exponential* performs poorly.  A *sine* scheduler is in line with such intuitions: due to the high temporal redundancy, many pixels have slight differences from the previous frames. In the beginning, we can take a large step with high certainty. In the last few steps, we have to cope with pixels of large entropy where we should take small steps to make more precise estimations. Increasing the number of steps can improve the performance, but for steps larger than 8, the gains are marginal and it will introduce computational overhead for training and inference.
>
> Further visualizations are provided in the Appendix, Section C. In Fig. 7, We can find that the model decodes images following an easy-to-hard process. Considering the high temporal redundancy of frame sequences, a large portion of pixels, mostly background, is easy to predict. The model starts from the tokens with the smallest entropy (e.g., sky and ground) to get context information. In the beginning,  we can take large steps because these tokens consume a little bitrate.  Gradually, when it "knows" more about the image, it refines the large-entropy tokens (e.g., the fast-moving billboard and horseshoe) which are hard to predict.   We should take a small step to make more accurate estimations gradually. This is analogous to the behaviors of a painter who starts with a sketch and then progressively refines it by filling in or tweaking the details.  This dynamic autoregressive strategy differs from the raster scanning order which might encounter a hard-to-easy dilemma and result in more bit-stream consumption.

---

> > ### Author Response · Authors · 2022-11-16
> > **Response to Reviewer 33TR [2/2]**
> >
> >
> > > Q3. Training details are missing, leading to poor reproducibility. Model details of how prior information is generated is missing too. Please clarify in appendix at the very least.
> >
> > We thank the reviewer for raising this point. We use some off-the-shell modules in our framework. To make it self-explained, we add more details in the revised paper.
> >
> > *In the Appendix, Section A:*
> >
> > The window size of W-MSA and SW-MSA is 4.  The number of attention heads is 8 for every attention layer.  We use two layers of MLP with input size,  hidden size, and output size all 768.
> >
> >
> > **Hyper-prior Encoder, Decoder.** The hyper-prior network compresses and stores a bitstream used as the hierarchical prior $\boldsymbol{s}_t$ (Ball ́e et al. 2018). The encoder downsamples the latent $\boldsymbol{y}_t$. At the receiver side,  the decoder recovers it by upsampling.
> >
> >
> > **ConvLSTM**. We use a ConvLSTM module (shi et al, 2015) to aggregate temporal priors of all previous frames  before $\boldsymbol{y}_{t-1}$. We  employ convolution with one-step LSTM.
> >
> > *Table 7. Hyper-prior encoder/decoder and ConvLSTM architecture. Conv(input\_channels, output\_channels, kernel\_size, stride, padding). ConvTranspose(input\_channels, output\_channels, kernel\_size, stride, padding, output\_padding). LSTM(input\_size, hidden\_size, num\_layers).*
> > | Hyper-prior Encoder                     | Hyper-prior Decoder                          | ConvLSTM                                            | ResBlock              |
> > | --------------------------------------- | -------------------------------------------- | --------------------------------------------------- | --------------------- |
> > | out: $\boldsymbol{z}_t$  (1, 96, 4, 4)  | out: $\boldsymbol{s}_t$ (1, 192, 16, 16)     | out: $\boldsymbol{r}_{t-1}$ (1, 256, 192)           | out: (1, 192, 16, 16) |
> > | Conv(96, 96, 5, 2, 2)                   | ConvTranspose(96, 192, 3, 1, 1, 0)           | $\boldsymbol{h}_{t-2}\rightarrow$ LSTM(192, 192, 1) |        skip connection : + in              |
> > | ReLU()                                  | ReLU()                                       | Reshape                                             |        ReLU()               |
> > | Conv(96, 96, 5, 2, 2)                   | ConvTranspose(96, 96, 5, 2, 2, 1)            | ResBlock                                            |        Conv(192, 192, 3, 1, 0)               |
> > | ReLU()                                  | ReLU()                                       | ReLU()                                              |         ReLu()              |
> > | Conv(192, 96, 3, 1, 1)                  | ConvTranspose(96, 96, 5, 2, 2, 1)            | Conv(192, 192, 3, 1,1)                              |          Conv(192, 192, 3, 1, 0)             |
> > | in: $\boldsymbol{y}_t$ (1, 192, 16, 16) | in: $\hat{\boldsymbol{z}}_{t}$ (1, 96, 4, 4) | in: $\boldsymbol{y}_{t-2}$ (1, 192, 16, 16)         | in: (1, 192, 16, 16)  |
> >
> >
> > **Multi-frame training.** We can apply multi-frame (up to 7 frames) and patch-size ($256\times 256$) for training. Using long video sequences for training can alleviate the accumulated errors propagating throughout inter-frame coding. In the first stage, we use two consecutive frames, including one I frame and one P frame,  to train our model for 1 M steps using the hyper-prior entropy model without MIMT. This stage targets a good image coder-decoder that can faithfully reconstruct the image with less emphasis on the bit rate cost. Then we add the MIMT entropy model to make a better bit-rate estimation and use two consecutive frames to minimize the rate-distortion loss for 1 M steps. Finally, we extend the length of the training video sequence to 7 frames for  300 K steps. The learning rate is set to 5e-5. We set the batch size as 8, using the Adam optimizer on a single V100 GPU.

---

### Official Review · Reviewer_gTZB · 2022-10-27

**Confidence:** 5
**Correctness:** 3
**Technical Novelty And Significance:** 3
**Empirical Novelty And Significance:** 3
**Recommendation:** 6

**Clarity, Quality, Novelty And Reproducibility:**

Idea is relatively clear, usage of the ConvLSTM was surprising in light of using a Transformer (another sequence to sequence model). Approach seems novel, but the many components might make it hard to reproduce.

**Strength And Weaknesses:**

## Strengths

- Great PSNR performance on common benchmark datasets, outperforming VTM in PSNR/RGB.
- Using Masked transformers makes sense from a compute perspective and is an interesting direction.
- Paper reports runtime numbers, important for this growing field.

## Weaknesses

Ablation Study and Clarity of Results. While the results are very convincing, I left the paper wondering what we actually learned. Which of the introduced components is how cruicial? In contrast to VCT, we have a) hyperprior, b) recurrenct prior, c) flow conditioning on E/D. While Table 4 answers some questions, in light of VCT I would have really like a row that is just "last two frames":

a) I am unconvinced that the ConvLSTM provides any benefit over just feeding the last two representations (i.e., why not feed y_{t-1}, y_{t-2} instead of y_{t-1}, r_{t-1}). The authors present no evidence that the (complicated and annoying to train) ConvLSTM provides any benefit.

b) Furthermore, and *more cruicially*, no ablation on c) flow conditioning is performed. I would be really interested in seeing how that affects PSNR.


**Summary Of The Paper:**

The paper proposes to use masked transformers in the representation space of a compressive autoencoder to do video coding. The masked transformer predicts the distribution of the representation auto-regressively, where more and more tokens are uncovered, similar to MaskGIT (Chang et al 2022). In contrast to other recent transformer work for video compression (VCT, Mentzer et al 2022), the transformer   does not just see see the last two representations (y_t-1, y_t-2), but also a hyper latent z_t, as well as the output of a ConvLSTM. Furthermore, the encoder and decoder transforms are conditioned on motion between adjacent frames, introducing again a predictive coding mode (in contrast to VCT): the frame at time step t depends on all reconstructions since the last I-frame.

**Summary Of The Review:**

I am intrigued by the rate distortion performance and using masking is elegant, but for acceptance we need more ablation studies to actually learn something from this paper besides "many components lead to good performance".

I am happy to change my rating if we get an ablation on the flow in the Encoder/Decoder (_b_ above), my concern about the ConvLSTM is less important (_a_ above) but that ablation would be nice to have.

---

> ### Author Response · Authors · 2022-11-16
> **Response to Reviewer gTZB [1/3]**
>
> We thank reviewer **gTZB**  for valuable feedback on our manuscript. We are happy that the reviewer appreciates the novel idea of using a masked transformer that makes sense from the decoding perspective and it is an interesting direction.
>
> > Q1.  Which of the introduced components is how cruicial? In contrast to VCT, we have a) hyperprior, b) recurrenct prior, c) flow conditioning on E/D. While Table 4 answers some questions, in light of VCT I would have really like a row that is just "last two frames":
>
> We thank Reviewer **gTZB** for helpful feedback on our work. The video compression framework, as the standard codec,  involves multiple elaborated components to compress the spatial-temporal redundancy from different perspectives.  The core idea of the proposed MIMT is underpinned by the masked transformer for entropy coding. To further improve the overall performance, we use hyper-prior, the last decoded frame, and recurrent prior as the inputs of the entropy model. The transformation coding is based on a flow-condiction encoder/decoder, following the good practice of contextual coding (Sheng et al., 2021; Li et al., 2021; 2022).
>
> To clarify the contribution of some key components, we have now added more ablation studies in the revised paper. Individual responses below are provided to address specific concerns.
>
> > Q2. I am unconvinced that the ConvLSTM provides any benefit over just feeding the last two representations (i.e., why not feed y_{t-1}, y_{t-2} instead of y_{t-1}, r_{t-1}). The authors present no evidence that the (complicated and annoying to train) ConvLSTM provides any benefit.
>
> We are very grateful for this suggestion from Reviewer gTZB.
>
> *In Appendix, Section B:*
>
> Exploiting inter-frame redundancy has always been a critical consideration for video compression. Generally, a deep learning model can use two consecutive frames $\\{y_\{t\}, y_\{t−1\}\\}$ (Lu et al., 2019; Li et al., 2021), three consecutive frames $\\{y_t, y_\{t−1\}, y_\{t−2\}\\}$ (Mentzer et al., 2022), four consecutive frames $\\{y_t, y_\{t−1\}, y_\{t−2\}, y_\{t−3\}\\}$ (Hu et al., 2021), or all previous frames using recurrent network (Ma et al., 2019; Yang et al., 2020). We use a one-step ConvLSTM to aggregate all previous frames before $y_{t−1}$. From Table 3, it can be discovered that recurrent prior $\boldsymbol{r}_{t−1}$ brings 11.1% bitrate savings.
>
> As the reviewer mentioned, one alternative to simplify the recurrent prior is to remove the ConvLSTM and use $\hat\{\boldsymbol\{y\}\}_\{t−2\}$ instead. An ablation study is conducted with results shown in Table 8. The second row shows that $\hat\{\boldsymbol\{y\}\}_\{t−2\}$ helps save significant bitrate. But there are additional $3.9\\%$ bitrate savings if we use recurrent prior $\boldsymbol\{r\}_\{t−1\}$. This result implies that $\hat\{y\}_\{t−1\}, \hat\{y\}_\{t−2\}$ contains most temporal context to compress $y_t$, but there is still room for improvement if we exploit a longer range of frames. This observation is consistent with the conclusion from FVC (Hu et al., 2021) and RLVC (Yang et al., 2020).
>
>
> *Table 8 Ablations on the recurrent prior $\boldsymbol{r}_{t-1}$ of MIMT.*
> | Inputs for Entropy Model                                                   | **UVG**  | MCL-JCV   | **HEVC-B** | Average   |
> | -------------------------------------------------------------------------- | -------- | --------- | ---------- | --------- |
> | $\boldsymbol\{s\}_t, \hat\{\boldsymbol\{y\}\}_\{t-1\}$                             | $+8.6\\%$ | $+11.3\\%$ | $+13.5\\%$  | $+11.1\\%$ |
> | $\boldsymbol\{s\}_t, \hat\{\boldsymbol\{y\}\}_\{t-1\}, \hat\{\boldsymbol\{y\}\}_\{t-2\}$ | $+2.2\\%$ | $+4.1\\%$  | $+5.4\\%$    | $+3.9\\%$  |
> | $\boldsymbol\{s\}_t, \hat\{\boldsymbol\{y\}\}_\{t-1\}, \boldsymbol\{r\}_\{t-1\}$       | $0\\%$    | $0\\%$     | $0\\%$      | $0\\%$     |
>
> [1] Guo Lu, Wanli Ouyang, Dong Xu, Xiaoyun Zhang, Chunlei Cai, and Zhiyong Gao. Dvc: An end-to-end deep video compression framework. In Proceedings of the IEEE/CVF Conference on Computer Vision and Pattern Recognition, pp. 11006–11015, 2019.
>
> [2] Jiahao Li, Bin Li, and Yan Lu. Deep contextual video compression. Advances in Neural Information Processing Systems, 34:18114–18125, 2021.
>
> [3] Fabian Mentzer, George Toderici, David Minnen, Sung-Jin Hwang, Sergi Caelles, Mario Lucic, and Eirikur Agustsson. Vct: A video compression transformer. arXiv preprint arXiv:2206.07307, 2022.
>
> [4] Siwei Ma, Xinfeng Zhang, Chuanmin Jia, Zhenghui Zhao, Shiqi Wang, and Shanshe Wang. Image and video compression with neural networks: A review. IEEE Transactions on Circuits and Systems for Video Technology, 30(6):1683–1698, 2019.
>
> [5] Ren Yang, Fabian Mentzer, Luc Van Gool, and Radu Timofte. Learning for video compression with hierarchical quality and recurrent enhancement. In Proceedings of the IEEE/CVF Conference on Computer Vision and Pattern Recognition, pp. 6628–6637, 2020

---

> > ### Author Response · Authors · 2022-11-16
> > **Response to Reviewer gTZB [2/3]**
> >
> >
> > > Q3. Furthermore, and more cruicially, no ablation on c) flow conditioning is performed. I would be really interested in seeing how that affects PSNR.
> >
> >
> >
> > Thank you for raising this point. Video compression methods heavily rely on flow-based prediction to reduce temporal redundancy in video sequences. We follow the good practice to use flow-based transformation coding (Sheng et al., 2021; Li et al., 2021; 2022).
> >
> > In the last row of Table 3, we conduct an ablation study on the image encoder by removing the temporal context.  The results show the contextual encoder provides significant bitrate saving of $+16.9\\%$ over the non-contextual encoder with independent image compression. This demonstrates that contextual coding is complementary to the proposed MIMT to boost the rate-distortion performance.
> >
> > For reference, the ablation study in DCVC (Li et al. 2021) verifies that the flow-based contextual coding can bring about $+12.9\\%$ bitrate savings for their model. The results also show, even when the model reduces to a non-contextual encoder/decoder similar to VCT, there is still a large margin of improvement over VCT.
> >
> >
> > *Table.  The last four columns show the BD rate increase over the anchor of the complete MIMT model.*
> >
> > | Ablation Options                                   | UVG       | MCL-JCV   | HEVC-B    | Average   |
> > | -------------------------------------------------- | --------- | --------- | --------- | --------- |
> > | non-contextual encoder/decoder                     | $+13.0\\%$ | $+18.4\\%$ | $+19.5\\%$ | $+16.9\\%$ |
> > | VCT  |  $+99.1\\%$         | $+75.4\\%$          |    NA     |   $+87.2\\%$        |
> >
> >
> >
> > **Reference:**
> >
> > Xihua Sheng, Jiahao Li, Bin Li, Li Li, Dong Liu, and Yan Lu. Temporal context mining for learned video compression. arXiv preprint arXiv:2111.13850, 2021.
> >
> > Jiahao Li, Bin Li, and Yan Lu. Deep contextual video compression. Advances in Neural Information Processing Systems, 34:18114–18125, 2021.
> >
> > Jiahao Li, Bin Li, and Yan Lu. Hybrid spatial-temporal entropy modelling for neural video compression. arXiv preprint arXiv:2207.05894, 2022.10

---

> > > ### Author Response · Authors · 2022-11-16
> > > **Response to Reviewer gTZB [3/3]**
> > >
> > >
> > > > Q4. Idea is relatively clear, usage of the ConvLSTM was surprising in light of using a Transformer (another sequence to sequence model). Approach seems novel, but the many components might make it hard to reproduce.
> > >
> > > Thank you for your valuable suggestions.  The proposed MIMT is built on a masked transformer encoder-decoder network and some other component that is off-the-shell and readily available. To make it clear and self-contained, we added more explanations in the revised paper.
> > >
> > > *In the Appendix, Section A:*
> > >
> > > The window size of W-MSA and SW-MSA is 4.  The number of attention heads is 8 for every attention layer.  We use two layers of MLP with input size,  hidden size, and output size all 768.
> > >
> > > **Hyper-prior Encoder, Decoder.** The hyper-prior network compresses and stores a bitstream used as the hierarchical prior $\boldsymbol{s}_t$ (Ball ́e et al. 2018). The encoder downsamples the latent $\boldsymbol{y}_t$. At the receiver side,  the decoder recovers it by upsampling.
> > >
> > >
> > > **ConvLSTM**. We use a ConvLSTM module (shi et al, 2015) to aggregate temporal priors of all previous frames  before $\boldsymbol{y}_{t-1}$. We  employ convolution with one-step LSTM.
> > >
> > > *Table 7. Hyper-prior encoder/decoder and ConvLSTM architecture. Conv(input\_channels, output\_channels, kernel\_size, stride, padding). ConvTranspose(input\_channels, output\_channels, kernel\_size, stride, padding, output\_padding). LSTM(input\_size, hidden\_size, num\_layers).*
> > > | Hyper-prior Encoder                     | Hyper-prior Decoder                          | ConvLSTM                                            | ResBlock              |
> > > | --------------------------------------- | -------------------------------------------- | --------------------------------------------------- | --------------------- |
> > > | out: $\boldsymbol{z}_t$  (1, 96, 4, 4)  | out: $\boldsymbol{s}_t$ (1, 192, 16, 16)     | out: $\boldsymbol{r}_{t-1}$ (1, 256, 192)           | out: (1, 192, 16, 16) |
> > > | Conv(96, 96, 5, 2, 2)                   | ConvTranspose(96, 192, 3, 1, 1, 0)           | $\boldsymbol{h}_{t-2}\rightarrow$ LSTM(192, 192, 1) |        skip connection : + in              |
> > > | ReLU()                                  | ReLU()                                       | Reshape                                             |        ReLU()               |
> > > | Conv(96, 96, 5, 2, 2)                   | ConvTranspose(96, 96, 5, 2, 2, 1)            | ResBlock                                            |        Conv(192, 192, 3, 1, 0)               |
> > > | ReLU()                                  | ReLU()                                       | ReLU()                                              |         ReLu()              |
> > > | Conv(192, 96, 3, 1, 1)                  | ConvTranspose(96, 96, 5, 2, 2, 1)            | Conv(192, 192, 3, 1,1)                              |          Conv(192, 192, 3, 1, 0)             |
> > > | in: $\boldsymbol{y}_t$ (1, 192, 16, 16) | in: $\hat{\boldsymbol{z}}_{t}$ (1, 96, 4, 4) | in: $\boldsymbol{y}_{t-2}$ (1, 192, 16, 16)         | in: (1, 192, 16, 16)  |

---

> > ### Comment · Reviewer_gTZB · 2022-11-18
> > **Response**
> >
> > Thanks for the response and further details. It is interesting to see how little the ConvLSTM adds over just using the last latent, and at the same time how much flow-based transforms help.
> >
> > I am happy to adapt my rating to borderline accept.

---

### Author Response · Authors · 2022-11-16
**Summary of revision**


We would like to thank all reviewers for their time and suggestions that help us improve our study. The most essential concerns from all reviewers are about the ablation studies. In response to feedback, we provide a general response here to points raised by multiple reviewers, individual responses below to address each reviewer’s concerns, and an updated manuscript.


1. Ablation on the flow-based contextual encoder/decoder. The flow-based prediction is a common practice in video compression, yet essential for rate-distortion performance. The proposed MIMT entropy model is assisted with off-the-shell flow-based contextual coding to achieve SOTA.
2. Ablation on MIMT with auto-regressive, and checkerboard entropy model. We make in-depth evaluations of different entropy models to demonstrate the benefits of MIMT.
3. The evaluation and motivation of the decoding scheduler. Apart from the default *sine* scheduler, we analyze two possible alternatives, i.e., linear and exponential.
4. Implementation details, including ConvLSTM, hyper-prior, etc.


In summary, the core idea of our study is a powerful masked transformer encoder-decoder network for spatial-temporal context modeling. We build our framework centric around the MIMT collaboratively with techniques of transformation coding and temporal context. We conduct a series of ablation studies to verify the effectiveness of components.

We appreciate all reviewers for their time and feedback, and we hope that our changes adequately address all concerns.

---

### Author Response · Authors · 2022-11-17
**Follow-up on Rebuttal**

Dear Reviewers,

Following your questions, we provided new ablation experiments to reveal the importance of the proposed MIMT entropy model. In addition, we improved the clarity of our method by adding more implementation details and discussions. We believe that the contents and the clarity of our paper are much improved in the revised version.

In light of this, we would like to know whether you believe these additional experiments and the updated manuscript have addressed your concerns.

Thank you for your time,

The Authors

---

### Public Comment · ~Donghui_Feng2 · 2023-05-08
**Doubt on Fig5, RD performance on HEVC-B dataset**

In Figure 5, RD performance on HEVC-B dataset (top-right), VTM and HM has almost the same performance. While in Table 1: BD-rate on HEVC-B,  the BD-rate of HM vs VTM is +40.4%.  Is Figure 5 correct? OR how is the BD-rate calculated?

![](https://notes.sjtu.edu.cn/uploads/upload_0e47a01332e9284d532a0985037bb9b2.png)

---

> ### Author Response · Authors · 2023-05-08
> **response on the HEVC-B dataset**
>
> Thank you for bringing this to our attention. We want to clarify that we have re-implemented the standard VTM and HM methods, considering the settings outlined in [ref1]. We can confirm that the BD-rate reported in Table 1 is accurate. However, upon further investigation, we have identified an error in the HEVC-B (PSNR) subfigure in Figure 5. We apologize for any confusion or inconvenience this may have caused.
>
> updated figure:
> https://drive.google.com/file/d/1Nd5CNi8T7ragST4CTQm1S_1QY7tarufR/view?usp=sharing
>
> [ref1] Li, J., Li, B. and Lu, Y., 2022, October. Hybrid spatial-temporal entropy modelling for neural video compression. In Proceedings of the 30th ACM International Conference on Multimedia (pp. 1503-1511).

---

> > ### Public Comment · ~Peng-Yu_Chen2 · 2023-05-23
> > **Concern about RD curves on UVG & MCL-JCV dataset**
> >
> > I would like to extend the discussion about RD performance in Figure 5.
> > Actually, I found that RD performance of `DMC` is **NOT** aligned to which reported in [ref 1] Figure 5.
> > As you updated the figure of RD curves on HEVC-B dataset, the performance of `DMC` can now match which reported in [ref 1].
> > In the mean time, the RD curve of `MIMT` is also updated.
> >
> > I wonder if there's similar problem on UVG & MCL-JCV dataset in Figure 5.
> > If so, could you update the results on these datasets too?
> >
> > [ref1] Li, J., Li, B. and Lu, Y., 2022, October. Hybrid spatial-temporal entropy modelling for neural video compression. In Proceedings of the 30th ACM International Conference on Multimedia (pp. 1503-1511).

---

### Decision · Program_Chairs · 2023-01-20

**Decision:**

Accept: notable-top-25%

**Justification For Why Not Higher Score:**

- All components of the proposed method except for masked image modeling were previously used in the context of neural compression.
- While the method achieves state-of-the-art results, it also consists of a large number of components which makes it less practical.

**Justification For Why Not Lower Score:**

- Domain highly relevant to ICLR, SOTA rate-distortion results on several benchmarks

**Metareview: Summary, Strengths And Weaknesses:**

The authors propose MIMT, a method for video compression, which builds on recent ideas in masked image modeling and entropy coding. In contrast to existing work where PMFs of tokens are auto-regressively predicted in raster-scanning order, MIMT can predict the PMFs in arbitrary order which is then used in an iterative decoding scheduler. When combined with other techniques such as motion compensation, hyper-prior modeling, and a recurrent prior the proposed method achieves state-of-the-art rate-distortion results and outperforms several recent engineered and neural codecs across a variety of established benchmark data sets and metrics. The reviewers find the work a solid contribution to the neural compression field and find that the ablations provided in the rebuttal sufficiently addressed the remaining major questions. Given the significance, novelty, clarity, and strong empirical performance, I will recommend acceptance of this manuscript. I ask the authors to submit a revised version adding the ablation studies provided in the rebuttal.

**Note From Pc:**

if the above contains the word "oral" or "spotlight" please see: "oral" presentation means -> notable-top-5% and "spotlight" means -> notable-top-25%. As stated in our emails, we are disassociating presentation type from AC recommendations